# Seasonal and diurnal variations of biogenic volatile organic compounds in highland and lowland ecosystems in southern Kenya

Yang Liu[1], Simon Schallhart[2], Ditte Taipale[3], Toni Tykkä[2], Matti Räsänen[3], Lutz Merbold[4,+], Heidi Hellén[2], Petri Pellikka[1,3]

[1] Department of Geosciences and Geography, University of Helsinki, P.O. Box 64, 00014 Helsinki, Finland
[2] Finnish Meteorological Institute, PL 503, 00101 Helsinki, Finland
[3] Institute for Atmospheric and Earth System Research / Physics, Faculty of Science, University of Helsinki, P.O. Box 64, 00014 Helsinki, Finland
[4] Mazingira Centre, International Livestock Research Institute, P.O. Box 30709, 00100 Nairobi, Kenya;
[+] now at Agroscope, Research Division Agroecology and Environment, Reckenholzstrasse 191, 8046, Zurich, Switzerland

**Correspondence:** Yang Liu (yang.z.liu@helsinki.fi)

**Abstract.** The East African lowland and highland areas consist of water-limited and humid ecosystems. The magnitude and seasonality of biogenic volatile organic compounds (BVOCs) emissions and concentrations from these functionally contrasting ecosystems are limited due to a scarcity of direct observations. We measured mixing ratios of BVOCs from two contrasting ecosystems, humid highlands with agroforestry and dry lowlands with bushland, grassland, and agriculture mosaics, during both the rainy and dry seasons of 2019 in southern Kenya. We present the diurnal and seasonal characteristics of BVOC mixing ratios and their reactivity, and estimated emission factors (EFs) for certain BVOCs from the African lowland ecosystem based on field measurements. The most abundant BVOCs were isoprene and monoterpenoids (MTs), with isoprene contributing > 70 % of the total BVOC mixing ratio during daytime, while MTs accounted for > 50 % of the total BVOC mixing ratio during nighttime at both sites. The contributions of BVOCs to the local atmospheric chemistry were estimated by calculating the reactivity towards the hydroxyl radical (OH), ozone ($O_3$), and the nitrate radical ($NO_3$). Isoprene and MTs contributed the most to the reactivity of OH and $NO_3$, while sesquiterpenes dominated the contribution of organic compounds to the reactivity of $O_3$.

The mixing ratio of isoprene measured in this study was lower than that measured in the relevant ecosystems in west and south Africa, while that of monoterpenoids was similar. Isoprene mixing ratios peaked daily between 16:00 and 20:00 with a maximum mixing ratio of 809 parts per trillion by volume (pptv) and 156 pptv in the highlands, and 115 pptv and 25 pptv in the lowlands, during the rainy and dry seasons, respectively. MT mixing ratios reached their daily maximum between midnight and early morning (usually 04:00 to 08:00) with mixing ratios of 254 pptv and 56 pptv in the highlands, and 89 pptv and 7 pptv in the lowlands, in the rainy and dry seasons, respectively. The dominant species within the MT group were limonene, α-pinene, and β-pinene.

EFs for isoprene, MTs, and 2-methyl-3-buten-2-ol (MBO) were estimated using an inverse modeling approach. The estimated EFs for isoprene and β-pinene agreed very well with what is currently assumed in the world's most extensively used biogenic emissions model, the Model of Emissions of Gases and Aerosols from Nature (MEGAN), for warm C4 grass, but the estimated EFs for MBO, α-pinene, and especially limonene, were significantly higher than that assumed in MEGAN for the relevant plant functional type. Additionally, our results indicate that the EF for limonene might be seasonally dependent in savanna ecosystems.

# 1 Introduction

Biogenic volatile organic compounds (BVOCs) are emitted from vegetation during, e.g., plant growth (e.g., Hüve et al.,
2007; Aalto et al., 2014; Taipale et al., 2020), reproduction (e.g., Andersson et al., 2002; Wright et al., 2005), and for defense (Niinemets, 2010; Holopainen and Gershenzon, 2010; Faiola and Taipale, 2020). The reactions of BVOCs with the hydroxyl radical (OH), nitrate radical ($NO_3$), and ozone ($O_3$) (Schulze et al., 2017; Ng et al., 2017) contribute to the oxidation capacity of the atmosphere (e.g., Mogensen et al., 2015), produce less volatile compounds which can form and growth atmospheric clusters (Matsunaga et al., 2005; Ehn et al., 2014; Kulmala et al., 2006), and impact cloud
condensation and scattering of solar radiation, affecting biosphere–atmosphere interactions and local/regional climate change (Claeys et al., 2004; Peñuelas and Staudt, 2010; Sporre et al., 2019) (Fig. 1).

Climate change affects BVOC emissions and oxidation through environmental conditions (Fig. 1: red arrows). Isoprene emissions are known to be both temperature and light dependent (Guenther et al., 1991, 1993; Wildermuth and Fall, 1996; Niinemets et al., 2004) and have been identified as the main contributor to increasing global BVOC levels in
response to global warming (Peñuelas and Staudt, 2010). Besides temperature and light, the emission of isoprene depends on soil water availability and thus responds to soil water stress (Guenther et al., 2012). The emission of monoterpenes is known to mainly be controlled by temperature, but the emission of certain monoterpenes (e.g., ocimene) depends greatly on the availability of light (Jardine et al., 2015; Guenther et al., 2012; Loreto et al., 1998). Mochizuki et al. (2020) estimated that monoterpene emissions will increase by 15 % with a 1 °C increase in air temperature due to climate
warming. The emission of certain monoterpenes is promoted by increasing soil moisture (Schade et al., 1999; Greenberg et al., 2012) and a decline in moisture–limited conditions (Bonn et al., 2019). Similar to isoprene and monoterpenes, 2-methyl-3-buten-2-ol (MBO) has shown that its emission is sensitive to light, temperature, and water stress (Gray et al., 2003). Increasing atmospheric carbon dioxide ($CO_2$) and air pollution (e.g., $O_3$) are also abiotic factors which affect BVOC emissions negatively or positively (Velikova, 2008; Masui et al., 2021). Since climate variability is rising
(Seneviratne et al., 2012), the emission of monoterpenes and isoprene is becoming more variable. This effect becomes especially pronounced in ecosystems that are vulnerable to climatic changes.

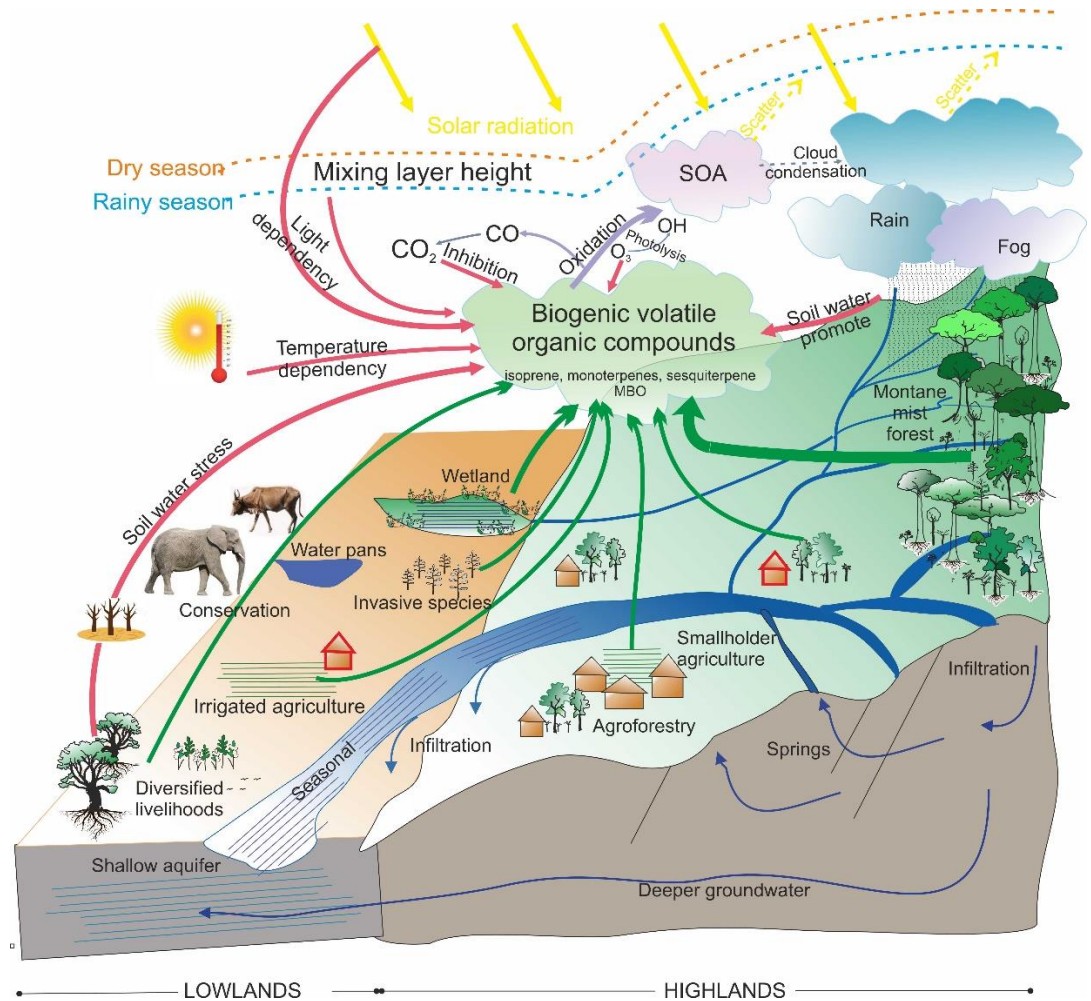

**Figure 1.** Atmospheric oxidation and abiotic effects on biogenic volatile organic compounds (BVOCs) in African highland and lowland ecosystems. The ecosystem impacts on $CO_2$ concentration (e.g. photosynthesis and respiration) are not shown in the figure above. SOA: secondary organic aerosol, OH: hydroxyl radical. The green arrows indicate BVOC sources from African ecosystems. BVOC oxidation and products are shown as purple arrows. The dashed yellow and purple arrows mean BVOC indirect impact to the climate. The abiotic effects on BVOC emissions are shown as red arrows. Blue arrows mean rivers and groundwater. The environments of our study areas are shown as the red house symbols. Figure courtesy: Gretchen Gettel, IHE Delft Institute for Water Education, Yang Liu and Petri Pellikka, University of Helsinki.

Dryland ecosystems and human–modified systems, including savannas, bushland, grassland, and agroforestry are more sensitive and vulnerable to ongoing climate change than other ecosystems (IPCC, 2014). It is estimated that around 18 % of global BVOCs are emitted from grass, shrubs, and crops (Guenther, 2013). This estimate is unfortunately connected with a large degree of uncertainty, since BVOC measurements from these ecosystems are rather scarce (e.g., Guenther et al., 2013). These climate–sensitive ecosystems are widely distributed and cover 55.2 % of tropical Africa (MDAUS BaseVue 2013, https://www.africageoportal.com/datasets/b4a808eba17d4294991880d9e120faee, last access: December 15, 2020), which have high potential on native ecosystem changes (Zabel et al., 2019), e.g. human–modified systems expansion at the expense of grassland and savannas, which can decrease the global BVOC levels (Unger, 2014). However, these aforementioned climate–sensitive ecosystems are also estimated to face a higher frequency of heat waves, hot nights, droughts, and flooding in the future climate (Niang et al., 2014; Kharin et al., 2018), which can promote or

inhibit the certain BVOC releases and make BVOC emissions more changeable. Models can simulate certain abiotic effects, for example temperature changes, soil water stress, and $CO_2$ inhibition, on BVOC emissions from these climate–sensitive ecosystems in current and future climate scenarios through the setting of suitable parameterizations, i.e., emission factors (EFs) and activity factors (Guenther et al., 2012; Emmerson et al., 2020). However, field measurements

focusing on volatile organic compounds from African ecosystems are very limited, especially on monoterpenoids (MTs), sesquiterpenes (SQTs), and MBO. Although previous BVOC measurements detected small quantities of MBO from African ecosystems (Jaars et al., 2016; Liu et al., 2021), MBO oxidation is an important source of ozone and hydrogen radicals (Steiner et al., 2007), which are both important oxidants for new particular formation in the local atmosphere (Jaoui et al., 2012; Zhang et al., 2014).

Previous measurements in tropical savannas have mainly focused on isoprene and/or monoterpenes (Guenther et al., 1996; Klinger et al., 1998; Greenberg et al., 1999, 2003; Otter et al., 2002; Harley et al., 2003; Stone et al., 2010; Jaars et al., 2016; Liu et al., 2021) (Fig 1: green arrows), and were measured during the local rainy season (except Jaars et al., 2016), which increases the challenge of BVOC estimation in these climate–sensitive African ecosystems.

Thus, the overall objective of this study was to quantify BVOC mixing ratios in the humid highland dominated by 95    agroforestry, and the dry lowlands with bushland and agriculture mosaic landscapes in Kenya during the rainy and dry season of 2019. We hypothesized significant differences in BVOC mixing ratios between land cover type, at diurnal scale and at season scale. We were interested in the diurnal as well as the seasonal variation in BVOC mixing ratios, and we estimated EFs for BVOCs to improve the representation of BVOC emissions from African ecosystems in models.

## 2 Material and methods

### 2.1 Experimental sites in Taita Taveta County

BVOC mixing ratios and meteorological measurements were set up in Taita Taveta County in southern Kenya. The county consists of dry savannas located in the lowlands between 500 and 1000 m a.s.l., and highlands ranging from approximately 1100 to 2200 m a.s.l. (Pellikka et al., 2018).

Taita Taveta County has two rainy and two dry seasons annually due to the Intertropical Convergence Zone forming a 105  bimodal rainfall pattern. The first rainy season (often referred to as the long rains) occurs between March and June, while the second rainy season (referred to as the short rains) is between October and December. The two rainy seasons are separated by dry seasons, with a short hot and dry season from January to February and a long cool and dry season from June to September (Ayugi et al., 2016; Wachiye et al., 2020). The highlands receive more rainfall than the lowlands. The annual precipitation is on average 1132 mm in Mgange (1768 m a.s.l.), corresponding to about twice the rainfall received 110  in Voi at 560 m a.s.l. (587 mm) (Erdogan et al., 2011). The annual temperature is 18.5 °C in Taita Research Station in the highlands and 22.3 °C in Maktau field site in the lowlands between 2013 and 2021. Both meteorological measurements are managed by the University of Helsinki, Finland. The length of sunlight remains $12 \pm 0.5$ hour through the entire year, with sunrise around $06:00 \pm 0.5$ hour and sunset about $18:00 \pm 0.5$ hour depending on the season (all times are given as East Africa Time, UTC +3).

The experimental sites were set up in the highlands in Wundanyi at Taita Research Station of the University of Helsinki and in the lowlands in the Maktau field site to represent the highland and lowland ecosystems, respectively (Fig. 2). The Taita Research Station (3° 40′ S, 38° 36′ E; 1415 m) is located in the middle of the Taita Hills on a windward slope. The landscape is characterized by small agricultural fields with a variety of crops, such as maize, beans, avocados, and grass,

with small native or exotic forest stands. The measurement station, which is fenced off, is surrounded by agroforestry
landscape, with the closest native and exotic forests at 200 m distance. The natural ecosystem of the Wundanyi site is
humid montane forest (Pellikka et al., 2009). Broadleaf evergreen trees and lush grass covered the ground layer during
the rainy season at the Wundanyi site (Fig. 2b), while part of the leaves were shed from trees and grass was dried out
around our instrument during the dry season (Fig. 2c). The Maktau field site (3° 25′ S, 32° 74′ E; 1056 m) is located in
the lowlands in which the natural ecosystem would be Acacia-commiphora bushland on savanna (Amara et al., 2020).
The measurement site is located inside a fenced farm growing maize, cassava, beans, and papaya trees, surrounded by
bushland. The soil on this site was not ploughed yet and the field was not sown or re-planted during our rainy season
measurements (Fig. 2d). The instrument was positioned near young cassava bush, with a distance of 50 m from the nearest
bushland edge. In the dry season, we collected the samples two weeks after the maize was harvested, the dry maize
residuals still remaining on the ground (Fig. 2e). The bushland surrounding the field was almost leafless during the dry
season sampling, while during the rainy season sampling, the new leaves were starting to sprout. The sites were chosen
for two reasons: 1) they represented typical highland agroforestry and lowland dry agriculture ecosystems with typical
bushland and forest cover, and 2) they provided safety and electricity for continuous measurements.

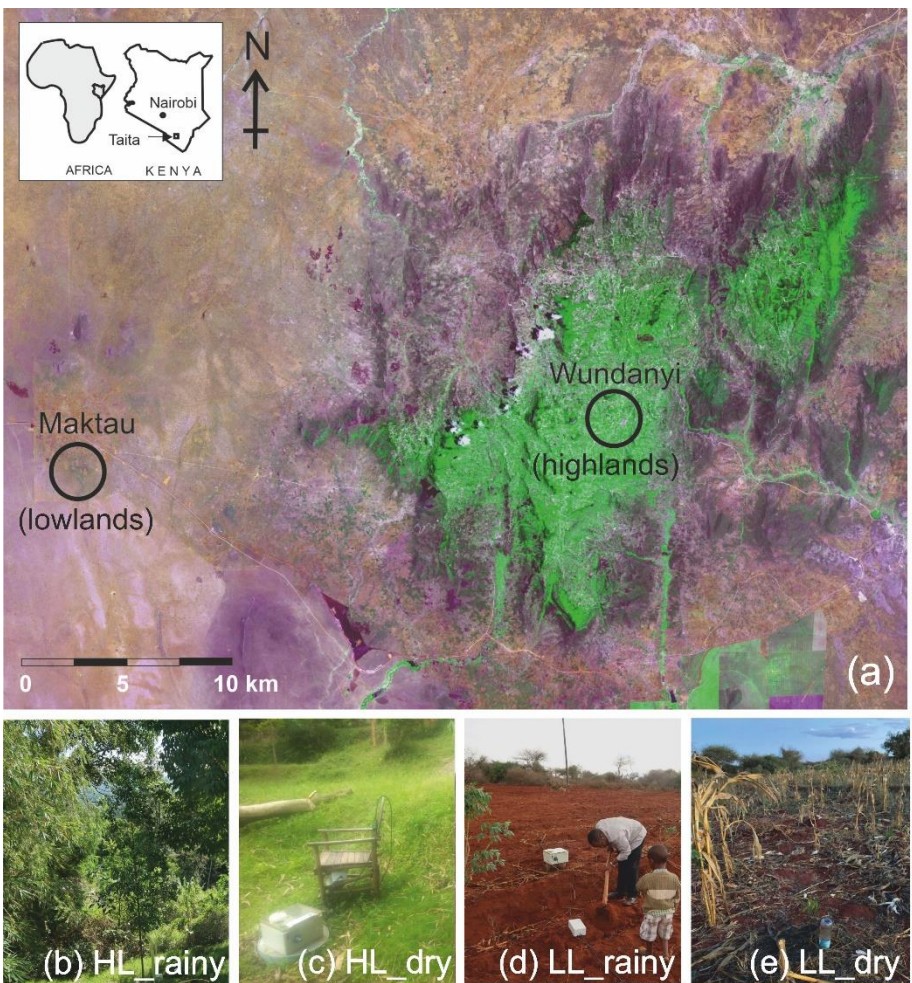

**Figure 2**. Locations of the highland site (HL) in Wundanyi and the lowland site (LL) in Maktau. Green color in the true
color Sentinel satellite image (a) shows the forests and agricultural area in Taita Hills, while magenta represents grassland
with a few fire scars. The brownish areas are areas with less land cover, such as dry bushland, dryland agriculture, and
areas used for livestock management. Photographs (b, c, d, e) show the phenological conditions and the surrounding
environments of the measurement sites during sampling in the rainy season in April and dry season in September.

**2.2 Sample collection and chemical analysis of BVOC mixing ratios**

We conducted four campaigns, each lasting several days, in the highlands and lowlands during the onset of the hot and long rainy season from April 10 to 17, 2019 and during the cool and long dry season from September 1 to 19, 2019 (Table A1).

The measurements took place upwind of the two sites, away from roads and at least 10 m away from the nearest residential buildings. Two autosamplers were used to collect air into thermal desorption sorbent tubes (STS 25,

PerkinElmer, Waltham, MA, USA), with a flow rate of 100 standard cubic centimeters per minute. All tubes were filled with Tenax TA (60/80 mesh, Sigma-Aldrich, St. Louis, MO, USA) and Carbopack B (60/80 mesh, Sigma-Aldrich, St. Louis, MO, USA). Although the cartridges were stored in ambient temperature during sampling, sorbents used in the tubes were hydrophobic and therefore water was not accumulated. In addition, tubes were flushed with helium for 5 minutes with the flow of 50 ml/min before desorption and analysis to remove traces of humidity.

The sampling time was generally 4 hours but was only 2 hours during the second campaign due to frequent power failures (Table A1). The sampling took place 25 cm above the ground so that flowing water during heavy rainfall events did not disturb the measurements. All samples were stored in the freezer (at approximately −15 °C) after collection (for 1 to 2 weeks) and before analysis (about 2 months). Tubes were stored in a closed box with ambient temperature and dark inside during the transportation to the Finnish Meteorology Institute (less than 1 week).

The mixing ratios of isoprene ($C_5H_8$), MBO ($C_5H_{10}O$), MTs ($C_{10}H_{16}$ and $C_{10}H_{18}O$), SQTs ($C_{15}H_{24}$), and bornyl acetate ($C_{12}H_{20}O$) were measured. MTs consisted of α-pinene, β-pinene, limonene, 3Δ-carene, ρ-cymene, camphene, terpinolene, linalool, and 1,8-cineol. SQTs consisted of longicyclene, iso-longifolene, β-caryophyllene, β-farnesene, and α-humulene. All samples were analyzed in the laboratory of the Finnish Meteorological Institute. An automatic thermal desorption device (PerkinElmer TurboMatrix 650) was connected to a gas chromatograph (PerkinElmer Clarus 600) with a DB-5MS

column (50 m × 0.25 mm, film 0.5 µm) and a mass-selective detector (PerkinElmer Clarus 600T). We desorbed all sample tubes at 300 °C for 5 minutes before cryo-focusing the samples in a Tenax TA cold trap (−30 °C) and injecting them into the column by rapidly heating the cold trap to 300 °C. The method, including potential losses, has been described in detail in Helin et al. (2020). The analytical uncertainties and the limit of quantification are shown in Table A2.

Standards in methanol solutions were used to calibrate the MBO, MTs, and SQTs. We injected the standards into the

sampling tubes and flushed away the methanol for 10 minutes before the analysis. The gaseous calibration standard (National Physical Laboratory) was applied for isoprene. Calibration samples were analyzed together with real samples.

**2.3 Complementary measurements and oxidant estimation**

**2.3.1 Meteorological data**

Meteorological data were measured simultaneously with sampling of BVOCs at Taita Research Station and Maktau

Weather Station. Hourly air temperature (CS215, Campbell Scientific, UK), relative humidity (CS215, Campbell Scientific, UK), precipitation (ARG100, EML, UK), wind speed and direction (Taita: Wind monitor 05103, R. M. Young, Traverse City, MI, USA; Maktau: 03002-L Wind Sentry Set, R. M. Young, Traverse City, MI, USA) were measured at both stations. All instruments were positioned at 1.5 m above the ground. Atmospheric pressure (CS106 Barometric pressure sensor, Vaisala, Finland), photosynthetic photon flux density (PPFD) (SKP215 Quantum, Skye Instruments,

UK), and soil moisture (CS650 sensor, Campbell Scientific, UK) were additionally measured at Maktau. PPFD sensor

was positioned around 4 m above the ground. Soil moisture was measured depths of 10 and 30 cm. Root–zone soil moisture calculation has been described in Räsänen et al. (2020).

The Chemistry Land–surface Atmosphere Soil Slab model was used to estimate mixing layer heights (MLHs) at the lowland site (Python version, Vilà-Guerau de Arellano et al., 2015). The model initial conditions were derived from the weather station observations. The sensible and latent fluxes from eddy covariance measurements were used as model input. These flux measurements were corrected by conserving the Bowen ratio using the net radiation measurements (Combe et al., 2015). The diurnal MLH data start from 06:00 and continue to 18:00, and the MLHs ranged from $337 \pm 25$ m to $2539 \pm 197$ m during the rainy season campaign in April, and from $361 \pm 18$ m to $2755 \pm 146$ m during the dry season campaign in September. All meteorology data during BVOC measurements are shown in Fig. 3.

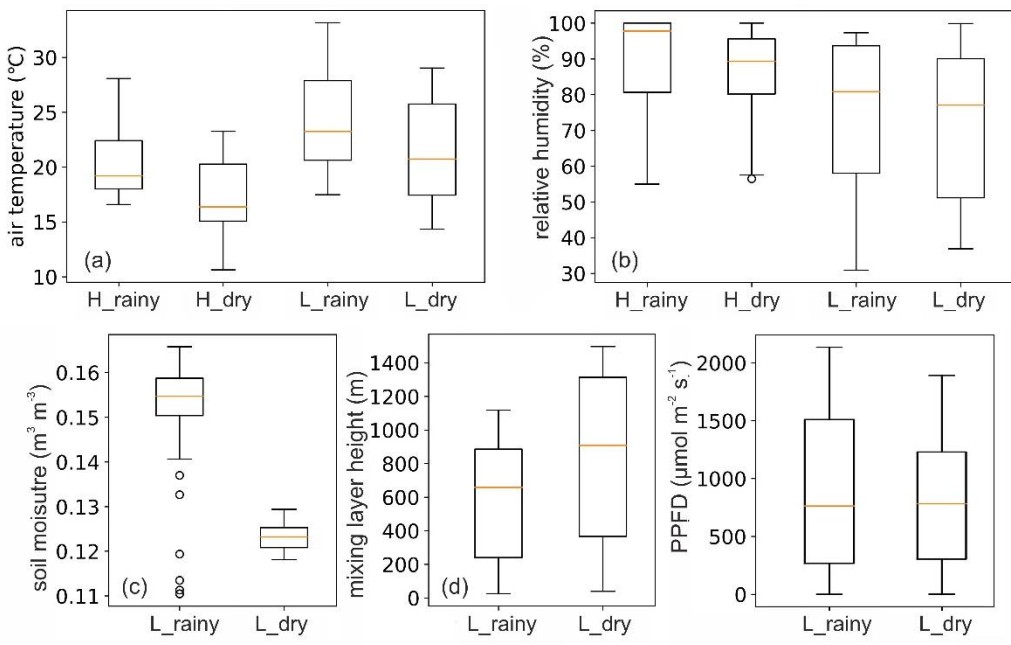

**Figure 3.** Meteorological measurements in the highland (H) and lowland (L) ecosystems during the rainy and dry seasons. PPFD = photosynthetic photon flux density.

### 2.3.2 Oxidant concentration estimation

Since the concentrations of oxidants were not measured directly during the campaigns, we used data observed by an Ozone Monitoring Instrument to acquire $O_3$ column densities to estimate surface $O_3$ concentrations (the conversion method is described at Ozonesonde, http://www.met.reading.ac.uk/~sws05ajc/teaching/ozonesonde.pdf, last access: August 23, 2021) and ultraviolet B (UVB) radiation intensity to calculate OH radical proxies using Eq. 1 (Rohrer et al., 2006; Petäjä et al., 2009).

$$OH_{proxy} = 5.62 \times 10^5 \times UVB^{0.62} \qquad (1)$$

The calculated average midday (local noon time) concentrations of $O_3$ were 31 parts per billion by volume (ppbv) and 29 ppbv in the rainy and dry seasons, respectively, while the corresponding concentration of OH was estimated to be $1.2 \times 10^6$ and $1.1 \times 10^6$ molecule cm$^{-3}$ in the rainy and dry seasons in our study area, respectively.

### 2.4 Reactivity calculation

Calculating the reactivity of BVOCs gives insight into the relative role of BVOCs in local atmospheric chemistry. The reactivity of BVOCs ($R_{i,x}$, where $i$ refers to the BVOC species and $x$ the oxidant species) was calculated by multiplying

the mixing ratio of a specific BVOC ($i$) with the corresponding reaction rate coefficient ($k_{i,x}$) of oxidants (including $O_3$, OH, $NO_3$) using Eq. 2.

$$R_{i,x} = BVOC_i \times k_{i,x} \qquad (2)$$

The parameter $k_{i,x}$ was calculated by using the average air temperature during each measurement (calculation equations described in Table A3). All of the reaction rate coefficients used in this study are provided in Table A4.

The atmospheric lifetime ($\tau$) of different BVOCs shows the oxidation speed of a specific compound or compound group in the atmosphere (Eq. 3). We calculated the lifetime of measured BVOCs in relation to $O_3$ and OH ($x$), as stated in Table A4.

$$\tau_{i,x} = \frac{1}{m}\sum_m (k_{i,x} \times Oxidant_x)^{-1} \qquad (3)$$

The amount of measurements in a certain period ($m$) was used to average over different measurement periods described hereafter as late night (00:00 to 04:00), early morning (04:00 to 08:00), late morning (08:00 to 12:00), early afternoon (12:00 to 16:00), late afternoon (16:00 to 20:00), and early night (20:00 to 00:00).

**2.5 Emission factor estimation**

EFs were estimated for isoprene, MBO, and the detected MTs using inverse modeling. In practice, a simple BVOC emissions and chemistry model was developed for this purpose. The model includes an emissions module based on Guenther et al. (2012). The emissions ($F_i$) of BVOCs ($i$) are calculated as Eq. 4:

$$F_i = \gamma_i \cdot EF_i \qquad (4)$$
$$\text{where } \gamma_i = C_{CE} LAI \gamma_{p,i} \gamma_{T,i} \gamma_{SM}$$

The $\gamma_i$ is an activity factor which accounts for emission responses due to various environmental parameters and phenological conditions. We considered BVOC emission responses due to light ($\gamma_{p,i}$, Eq. 5), temperature ($\gamma_{T,i}$, Eq. 6), and soil moisture ($\gamma_{SM}$, Eq. 7). A value of 0.57 was assigned to the canopy environment coefficient ($C_{CE}$) (Simpson et al., 1999, 2012; Guenther et al., 2012), while the one-sided leaf area index (LAI, obtained from PROBA-V, spatial resolution 300 m, https://land.copernicus.eu/global/products/lai, last access: December 14, 2020) was kept constant at a value of 1.53 $m^2$ $m^{-2}$ (April) or 0.3 $m^2$ $m^{-2}$ (September).

$$\gamma_{p,i} = (1 - LDF_i) + LDF_i \times C_p[(\alpha \times PPFD)/((1 + \alpha^2 \times PPFD^2)^{0.5})] \qquad (5)$$
$$\text{where } C_p = 0.0468 \times \exp(0.0005 \times [P_{24} - P_s]) \times [P_{240}]^{0.6}$$
$$\alpha = 0.004 - 0.0005 \ln(P_{240})$$

The parameter $LDF_i$ is the light dependent fraction of the emission of each individual BVOC and the values are provided in Guenther et al. (2012) Table 4. $P_s$ is the standard condition for PPFD, averaged over the past 24 h, and was set to 200 µmol $m^{-2}$ $s^{-1}$ (Guenther et al., 2012). $P_{24}$ and $P_{240}$ are the average PPFD of the past 24 h and the past 240 h, respectively.

$$\gamma_{T,i} = (1 - LDF_i) \times \exp(\beta_i(T - T_s)) + LDF_i \times E_{opt} \times \left[ C_{T2} \times \frac{\exp(C_{T1} \times x)}{C_{T2} - C_{T1} \times (1 - \exp(C_{T2} \times x))} \right] \quad (6)$$

$$\text{where } E_{opt} = C_{eo,i} \times \exp(0.05 \times (T_{24} - T_s)) \times \exp(0.05 \times (T_{240} - T_s))$$

$$x = \left[ \left( \frac{1}{313 + (0.6 \times (T_{240} - T_s))} \right) - \left( \frac{1}{T} \right) \right] / 0.00831$$

$C_{eo,i}$ is an emission-class dependent empirical coefficient for each BVOC in (Table 4 in Guenther et al., 2012). $T_s$
represents the standard conditions for leaf temperature and is equal to 297 K. $T_{24}$ and $T_{240}$ are the average leaf temperature
of the past 24 h and the past 240 h, respectively. The leaf temperatures were calculated from observed air temperatures
(Eqs 14.2 to 14.6 in Campbell et al., 1998). $\beta_i$, $C_{T1,i}$ and $C_{T2}$ are the empirically determined coefficients. We used 230
for $C_{T2}$, according to Guenther et al. (2012), and values for $\beta_i$ and $C_{T1,i}$ from Table 4 in Guenther et al. (2012).

$$\gamma_{SM,isoprene} = \begin{cases} 1 & \theta > \theta_l \\ \frac{\theta - \theta_w}{\Delta\theta_l} & \theta_w < \theta < \theta_l \\ 0 & \theta < \theta_w \end{cases} \quad (7)$$

where $\theta$ is the volumetric water content of soil. $\theta_l = \theta_w + \Delta\theta_l$, $\theta_w$ is wilting point and was set to 0.1 m$^3$ m$^{-3}$ for the
Maktau site (Räsänen et al., 2020), while $\Delta\theta_l$ is an empirical parameter which equals 0.04 (Guenther et al., 2012). $\gamma_{SM}$ is
only applied for estimation of the emission of isoprene according to Guenther et al. (2012).

The model's chemistry module consists of the first step in the oxidation of the BVOCs by O$_3$ and OH using the
reaction rate coefficients listed in Table A3. Reactions with NO$_3$ were omitted, because simulations were only carried out
using daytime observations. The model takes the following parameters as input: observations of PPFD, air temperature,
and soil moisture from the Maktau site, estimated leaf temperatures, estimated concentrations of O$_3$ and OH (Section
2.3.2), modeled daytime MLHs (Section 2.3.1), and LAI. In the model, the concentration of O$_3$ is kept constant within a
day, while the daily pattern of the OH concentration follows the solar zenith angle.

Initial estimations were made for the EFs, and the mixing ratios of the BVOCs were predicted using the model for
one campaign day at a time. The predicted and measured daytime BVOC mixing ratios were then compared (2–5 data
points per day) and the sum of the squared differences between the predicted and observed mixing ratios was calculated
for each individual BVOC for each day. A new estimation for the values of the EFs was made and the process was iterated
until a minimum sum of the squared differences was obtained (Table A5). The EF, for each individual BVOC, which led
to this minimum value, was considered the most appropriate value for the EF for that particular day (Table A5). Similar
simulations were conducted for each measured day. The median values of the estimated EFs during either the rainy or
dry season, for each individual BVOC, are our best estimates for the BVOC EFs for the agriculture site located in the
savanna ecosystem at Maktau field site. A comparison of the modelled and measured BVOC concentrations, which lead
to the minimum sums of the squared differences between the predicted and observed mixing ratios, is provided in Figs.
A1−1 and A1−2. Similar estimations of EFs were not conducted for the highland site, due to lack of necessary input data
to the model.

**3 Results and discussion**

**3.1 Seasonal and diurnal variations of BVOC mixing ratios**

For most of the compounds studied, the daily mean mixing ratio was higher during the rainy season than during the dry season. In the highlands, the daily mean isoprene mixing ratio ranged from 134 to 442 pptv in the rainy season, and ranged from 36 to 150 pptv in the dry season. The daily mean mixing ratio of MTs was 117 to 233 pptv in the rainy season, and was 8 to 75 pptv in the dry season. And that of SQTs was 2 to 30 pptv in the rainy season and 1 to 3 pptv in the dry season. In the lowlands, the daily mean mixing ratios of isoprene ranged from 22 to 69 pptv and from 6 to 15 pptv, in the rainy and the dry season, respectively. The mixing ratio of MTs were from 29 to 96 pptv in the rainy season, and from 3 to 9 pptv in the dry season. For SQTs, the daily mean mixing ratios ranged from 1 to 2 pptv and was less than 1 pptv, in the rainy and the dry season, respectively.

### 3.1.1 Mixing ratios of isoprene and monoterpenoids

Isoprene and MTs explained over 88 % of the total BVOC mixing ratios of all collected samples, and their mixing ratios in the rainy season were higher than in the dry season in both the highlands and lowlands. The seasonal mean ± standard deviation of the isoprene mixing ratio was 252.2 ± 285 pptv and 66.6 ± 75 pptv in the highlands in the rainy and the dry season, respectively, while the corresponding values were 145.5 ± 73 pptv and 35.2 ± 42 pptv for MTs (Fig. 4). In the lowlands, the mixing ratio of isoprene was 55.3 ± 56 pptv and 11.2 ± 9 pptv in the rainy and the dry season, respectively, while the corresponding values for MTs were 57.8 ± 46 pptv and 4.1 ± 4 pptv. Isoprene and all the MTs showed a clear mixing ratio maximum in the rainy season, and the seasonal mixing ratios of isoprene and MTs remained lower in the lowlands than in the highlands. The temporal variability of measured BVOCs presented in Fig. A2.

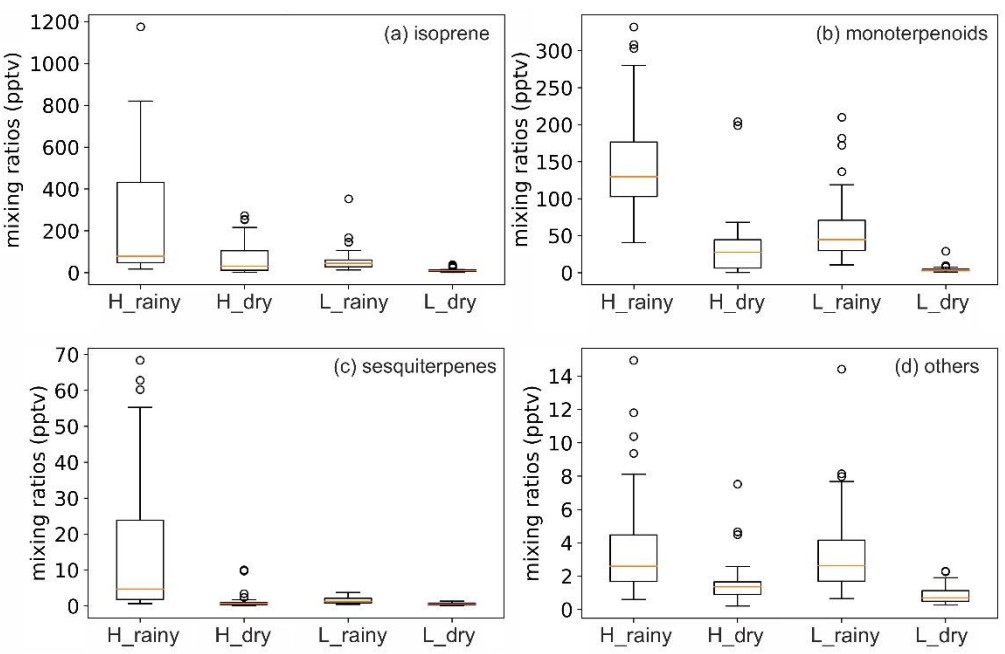

**Figure 4.** Mixing ratios of isoprene, monoterpenoids, sesquiterpenes, and others (2-methyl-3-buten-2-ol and bornyl acetate) in the highland and lowland ecosystems during the rainy and dry seasons.

Significantly higher temperature, more emitters from different vegetation types, as well as the lower mixing layer heights were probably the main factors promoting higher mixing ratio during the rainy season (Fig. A3, Table 1). Soil moisture was additionally so low during the dry season (Figs. A4g, h) that it has most probably reduced the emission rate of isoprene. PPFD, RH and estimated atmospheric oxidant concentrations stayed largely the same during the two seasons and thus did not influence the concentration difference. The slightly different PPFD between the two seasons should not

have impacted the emission of isoprene, since the light conditions during both seasons were still higher than the saturation point for the production and emission of isoprene (Figs. A4f, m; Guenther et al., 2006). It is likely that the significantly higher LAI in the highlands also caused higher BVOC mixing ratios in the highlands compared to the lowlands. Additionally, the vegetation type in the two ecosystems are different, which might also contribute to the difference, though in which direction is unclear, since emission rates from the particular plant species populating the areas have not been reported so far to our knowledge.

**Table 1**. LAI, the concentration of ozone, and meteorology conditions detected or estimated in the highland and the lowland sites during the campaigns. Parameters which were not measured in situ are indicated.

| Season | Site | $LAI^a$ $(m^2\ m^{-2})$ | $T^b$ (°C) | $T\_Dt^c$ (°C) | $PPFD\_Dt^d$ $(\mu mol\ m^{-2}\ s^{-1})$ | $RH^e$ (%) | $MLH\_Dt^f$ (m) | $SM^g$ $(m^3\ m^{-3})$ | $O_3^h$ (ppb) |
|---|---|---|---|---|---|---|---|---|---|
| Rainy | Highland | 2.08 | 20.9 | 22.7 | nan | 87.3 | nan | nan | 30.9 |
|  | Lowland | 1.53 | 23.9 | 25.5 | 867.7 | 75.1 | 663.9 | 0.15 | 31.2 |
| Dry | Highland | 1.9 | 17.3 | 18.6 | nan | 86.8 | nan | nan | 28.6 |
|  | Lowland | 0.3 | 20.5 | 22.2 | 865.5 | 75.0 | 819.6 | 0.12 | 28.8 |

[a] LAI: leaf area index (from satellite); [b] T: daily mean ambient temperature; [c] T_Dt: average of daytime (6 a.m. to 18 p.m.) ambient temperature; [d] PPFD_Dt: daytime mean photosynthetic photon flux density; [e] RH: daily mean relative humidity; [f] MLH_Dt: average of daytime mean mixing layer height (estimated); [g] SM: daily mean soil moisture; [h] $O_3$: estimated daily mean concentration of Ozone. 'nan': no observations available.

The mixing ratio of isoprene showed distinct diurnal variation in the highlands during both the rainy and dry seasons, but in the lowlands only during the dry season (Figs. A5−1 to A5−4). The mixing ratio of isoprene increased in the morning, coinciding with sunrise, and stayed high during the rest of the day. The measured mixing ratio of isoprene contributed on average 37 % and 84 % in the highlands and lowlands, respectively, to the total BVOC mixing ratio (Fig. 5).

The mixing ratios of MTs showed higher mixing ratios during night and dark hours than during light hours, particularly during the dry season in both ecosystems (Figs A5−1 to A5−4). Higher mixing ratios of MTs during night have been observed earlier in savannas in South Africa (Gierens et al., 2014), and needleleaf forest in California, and Finland (Bouvier-Bown et al., 2009; Hakola et al., 2012). Even though the MT emissions are expected to be highest during daytime, the mixing ratio of MTs is lower since the mixing, and therefore dilution, is highest during daytime and lowest during the night (Mogensen et al., 2011; Hellén et al., 2018).

The diurnal maximum mixing ratio of MTs was on average 254 and 56 pptv in the highlands, and 89 and 11.5 pptv in the lowlands, in the rainy and the dry seasons, respectively. The diurnal variations of α-pinene and limonene controlled the changes in total MT mixing ratio and contributed over 60 % to the total MT mixing ratio. Decreasing mixing ratios of limonene between day and night led the diurnal variation of the total mixing ratio of MTs in the rainy season, while decreasing α-pinene controlled the diurnal variation of total MTs in the dry season. The minimum diurnal mixing ratio of MTs occurred in the early night during the rainy season and around noon in the dry season.

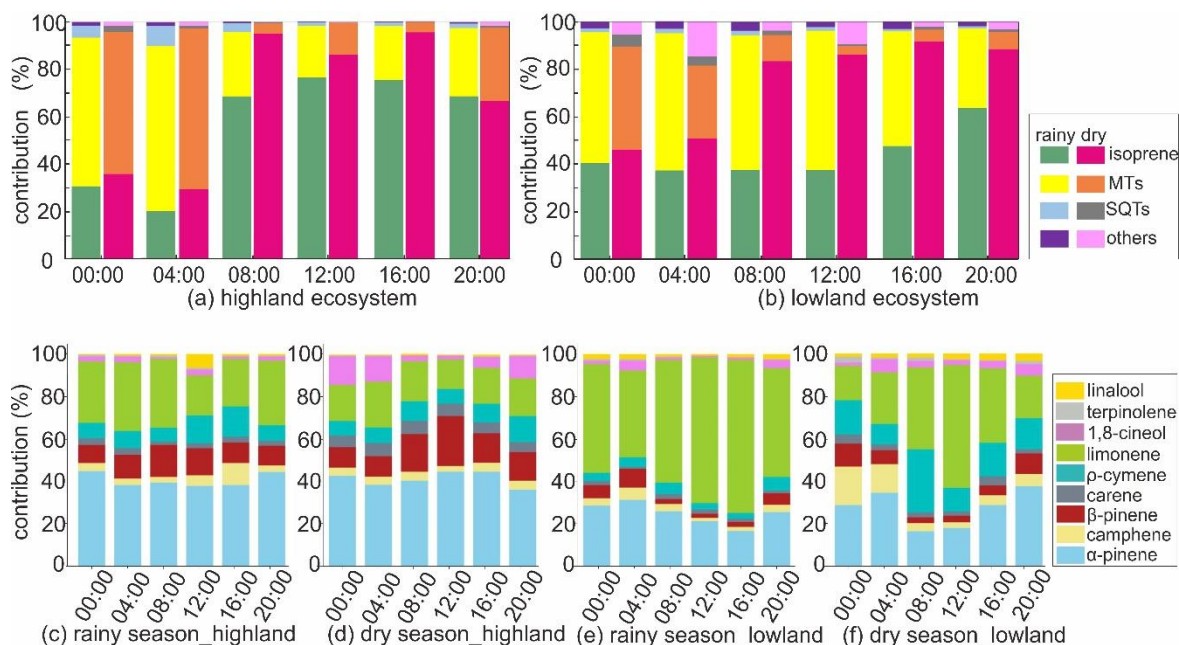

**Figure 5.** Diurnal contribution of biogenic volatile organic compounds **(a, b)** and contribution of monoterpenoids (MTs) **(c, d, e, f)** across the highland site and the lowland site in the rainy and the dry seasons. SQTs = sesquiterpenes, others = 2-methyl-3-buten-2-ol and bornyl acetate.

The isoprene mixing ratio ranged from 730 to 1820 pptv in the rainy season of 1996 in a tropical forest in northern Congo, which was covered by evergreen or semi-evergreen trees (Serça et al., 2001). A similar level of isoprene mixing ratio was observed in a forest ecosystem near Enyela, northern Congo, with values ranging from 700 to 1000 pptv at the end of the rainy season of 1996 (Greenberg et al., 1999). In western Africa, isoprene mixing ratios of over 1000 pptv during daylight hours were measured in a forest surrounded by a woodland savanna ecosystem in Benin (Saxton et al., 2007). The western Africa and the two central Africa measurements aforementioned all showed at least an order of magnitude higher isoprene mixing ratios compared with the measurements in the highlands (Wundanyi) of this study (Table 1). The measured mixing ratios of α-pinene, limonene, and β-pinene in Wundanyi were comparable to the corresponding compound levels from the aforementioned forest measurements. The mixing ratios of isoprene and MTs in Wundanyi are comparable to our previous measurements from three types of montane native forests of Taita Hills in southern Kenya (Liu et al., 2021). The mixing ratios of isoprene and MTs at the lowland site in Maktau were about four times lower than the corresponding levels measured from savanna ecosystems in Central Africa Republic (Boali) and South Africa (Greenberg et al., 1999; Harley et al., 2003), and grass, shrubland in western Senegal (Grant et al., 2008), and considerably lower than the corresponding compound levels from woodland in Botswana (Greenberg et al., 2003). The mixing ratios of isoprene and limonene in the rainy season in Maktau are higher than the levels of the corresponding compounds in grassland in Welgegund, South Africa, while the mixing ratios of α-pinene and β-pinene, both in the rainy and the dry seasons, as well as isoprene and limonene in the dry season in Maktau, were lower than the values reported by Jaars et al. (2016). The mixing ratios of α-pinene, limonene, and β-pinene in the rainy season in Maktau were all in the range of the mixing ratios of the corresponding compounds in our previous measurements, while that of isoprene was at lower levels than previously reported (Liu et al., 2021). The differences in mixing ratios between our measurement and from these aforementioned studies could be affected by several factors, e.g. dominant plant species and their distribution, temperature and light, wind speed/direction, mixing layer height, etc. But we were not able to find the key reasons based on the limited details from the other sites.

**Table 2.** Mixing ratios of biogenic volatile organic compounds in different ecosystems in Africa (mixing ratios are presented as median and mean values, except those with extra explanations, e.g., "midday," "minimum/maximum", 'mean $\pm$ STD'. The unit of mixing ratios are presented in part per trillion, pptv).

| Location | Time | Vegetation | Compound | Mixing ratio median (mean) | Reference |
|---|---|---|---|---|---|
| Wundanyi, Kenya (38.4° E, 3.4° S) | April and September 2019 | Agroforestry | Isoprene | Rainy: 78 (252) Dry: 34 (66) | This study |
| | | | α-Pinene | Rainy: 54 (59) Dry: 12 (15) | |
| | | | Limonene | Rainy: 37 (42) Dry: 3.4 (6.4) | |
| | | | β-Pinene | Rainy: 13 (15) Dry: 3.9 (4.7) | |
| Maktau, Kenya (32.7° E, 3.3° S) | | Savanna bushland | Isoprene | Rainy: 43 (55) Dry: 8.7 (11) | |
| | | | α-Pinene | Rainy: 8.0 (14) Dry: 0.7 (1.1) | |
| | | | Limonene | Rainy: 27 (34) Dry: 1.0 (1.2) | |
| | | | β-Pinene | Rainy: 1.6 (2.5) Dry: 0.2 (0.3) | |
| Enyele, Democratic Republic of the Congo (18° E, 3° N) | November and December 1996 | Forest | Isoprene | 700 to 1000 | Greenberg et al., 1999 |
| | | | α-Pinene | 30 to 100 | |
| Boali, Central African Republic (18° E, 4.5° N) | | Savanna | Isoprene | 100 to 400 | |
| | | | α-Pinene | 20 to 30 | |
| Northern Congo (16.2° E, 2.1° S) | March 1996 | Tropical evergreen forest, semi-evergreen forest | Isoprene | mean $\pm$ STD: 1820 $\pm$ 870 | Serça et al., 2001 |
| | November 1996 | | Isoprene | mean $\pm$ STD: 730 $\pm$ 480 | |
| | March and November 1996 | | β-Pinene | < 10 | |
| South Africa (29.8° E, 25.0° S) | February 2001 | *Combretum–Acacia* savanna | Isoprene | Midday 390 | Harley et al., 2003 |
| Botswana (23.3° E, 19.5° S) | February 2001 | Mopane woodland | α-Pinene | Minimum < 1000 Maximum > 2000 | Greenberg et al., 2003 |

| Location | Date | Vegetation type | Compound | Value | Reference |
|---|---|---|---|---|---|
| Benin (1.4° E, 9.4° N) | June 2006 | Forest | Isoprene | Day_maximum > 1000 Night_maximum > 500 | Saxton et al., 2007 |
| | | | Limonene | Few tens to 5000 | |
| Republic of Senegal (17.1° W 14.7° N) | September 2006 | Grasses, shrubs | Isoprene | Minimum 200 Maximum 400 | Grant et al., 2008 |
| Benin (2.7° E, 10.1° N) | August 17, 2006 | Subtropical forest | Isoprene | Midday 1184 mean ± STD: 294 ± 333 | Stone et al., 2010 |
| Welgegund, South Africa (26.9° E, 26.6° S) | February 2011 to February 2012 (1st); & December 2013 to February 2015 (2nd) | Grassland | Isoprene | 1st:14 (28), 2nd: 14 (23) | Jaars et al., 2016 |
| | | | α-Pinene | 1st: 37 (71), 2nd: 15 (57) | |
| | | | Limonene | 1st: 21 (30), 2nd: 16 (54) | |
| | | | β-Pinene | 1st: 9 (19), 2nd: 3 (5) | |
| Kenya (32 to 38° E, 3.2 to 3.4° S) | April 2019 | Montane forest | Isoprene | 741 (706) | Liu et al., 2021 |
| | | | α-Pinene | 74 (75) | |
| | | | Limonene | 6.6 (7.7) | |
| | | | β-Pinene | 7.2 (7.9) | |
| | | Grass and shrubs | Isoprene | 735 (713) | |
| | | | α-Pinene | 30 (25) | |
| | | | Limonene | 50 (56) | |
| | | | β-Pinene | 5.3 (4.6) | |

### 3.1.2 Mixing ratios of sesquiterpenes, MBO, and bornyl acetate

The mixing ratios of SQTs were low and contributed to around 3 % of the total BVOC mixing ratios in all samples. SQTs showed seasonal and diurnal variations similar to those of MTs, but their mixing ratio was much lower than that of MTs, with seasonal mean SQT mixing ratios of 15.0 ± 19 pptv and 1.1 ± 2 pptv in the highlands, and 1.5 ± 0.9 pptv and 0.5 ± 0.3 pptv in the lowlands, in the rainy and the dry seasons, respectively. SQTs are very reactive and therefore their contribution to the local atmospheric chemistry can still be significant. The highest daily means were measured during the nighttime, which was the same as in the case of the MTs. β-caryophyllene showed the highest mixing ratios among the SQTs, followed by β-farnesene and/or α-humulene measured in both the rainy and dry seasons. The diurnal trend of β-caryophyllene and β-farnesene followed the variation of total SQTs.

The mixing ratios of MBO and bornyl acetate were both low. MBO explained 2.6 % of the total BVOC mixing ratio of all samples, while bornyl acetate explained 0.5 %. Both compounds have seasonal and diurnal variations. The seasonal mean mixing ratios of MBO and bornyl acetate were 5 and 1.5 times higher in the rainy season than in the dry season in the highlands, respectively, and the mixing ratios of both BVOCs were 6 times higher in the lowlands. The diurnal mean mixing ratios of MBO and bornyl acetate were around 4 and 0.8 pptv in the rainy season in both the highlands and lowlands. MBO mixing ratios were 1 and 0.7 pptv in the dry season in the highlands and lowlands, while that of bornyl

acetate was 0.6 and 0.1 pptv, respectively. The daily mean mixing ratio of bornyl acetate was lower than 1 pptv in the rainy and dry seasons both in the highlands and lowlands.

Jaars et al. (2016) measured MBO for the first time in Africa, and they reported that the mean mixing ratios of MBO were 12 pptv and 8 pptv in their first and second campaign, respectively, which are higher than the mean MBO mixing ratios measured in the highlands and lowlands in this study. Guenther (2013) stated that MBO is emitted from most isoprene–emitting vegetation at an emission rate of $\sim$ 1 % of that of isoprene. The Welgegund data (Jaars et al., 2016) showed that MBO is approximately 30 % of the isoprene mixing ratio, and thus their study indicated that MBO at

Welgegund is most likely from other MBO emitting species than from isoprene emitters. MBO are higher than 1 % of isoprene mixing ratios in our study, which was 3.7 % and 6.3 % of the isoprene mixing ratio in the highlands in the rainy and dry seasons, respectively, and 7.6 % and 9.8 % in the lowlands. Unfortunately, we could not partition the source of MBO emitter(s) in this study area during our measurements.

    Be aware that no measurements were conducted during the short hot (January to February) and short cool (October to

December) season, and it is likely that the mixing ratios of BVOCs are different during those seasons than what is presented here, due to differences in e.g. environmental conditions and phenology status.

## 3.2 Reactivity of the measured BVOCs with oxidants

The reactivity toward $O_3$, OH, and $NO_3$ was calculated using the measured BVOC mixing ratios (Fig. 6). The $O_3$ reactivity of SQTs was 5 to 30 times higher than for other BVOCs, with β-caryophyllene having the highest contribution to the total

$O_3$ reactivity. The strong relative importance of the SQTs compared with other BVOCs for the local $O_3$ reactivity has also been seen in the ambient air of a Scots pine forest in Finland (Hellén et al., 2018). Out of the total BVOCs, MTs contributed most to the $NO_3$ reactivity, an average of 13 and 15 times more than isoprene and SQTs, respectively. MTs also contributed to the OH reactivity, with a 0.7 to 1.9 times higher contribution than isoprene during nighttime, while isoprene is the dominant BVOC contributor to the OH reactivity during the day, with 3.1 to 3.5 times higher contributions

than MTs.

    Isoprene shows the highest mixing ratio of BVOCs in this study. The atmospheric lifetime of isoprene is 34 hours and 2.3 hours with $O_3$ and OH, respectively. Follow that of isoprene, limonene ($\sim$ 2 hours) and α-pinene ($\sim$ 4 hours) have higher mixing ratios, and are detected to have a relatively short lifetime with OH and $O_3$ compared with other MTs (except terpinolene and linalool). Higher importance of limonene and α-pinene for OH reactivity than other MTs was also

observed in a savanna ecosystem in South Africa (Jaars et al., 2016), which reported that both compounds also had higher mixing ratios than other MTs during their campaigns. Compared with other MTs, limonene has a significantly higher yield for highly oxygenated organic molecules (Ehn et al., 2014; Bianchi et al., 2019), which has been found to be a major component of secondary organic aerosols (e.g., Ehn et al., 2014; Mutzel et al., 2015), for which higher limonene is expected to have a strong impact on local aerosol production in southern Kenya as well. The low mixing ratios of β-

caryophyllene and α-humulene have shorter lifetimes with OH and $O_3$ than other SQTs and BVOCs. The lifetimes of β-caryophyllene and α-humulene are a few minutes with $O_3$ and about 1 hour with OH (Table A4).

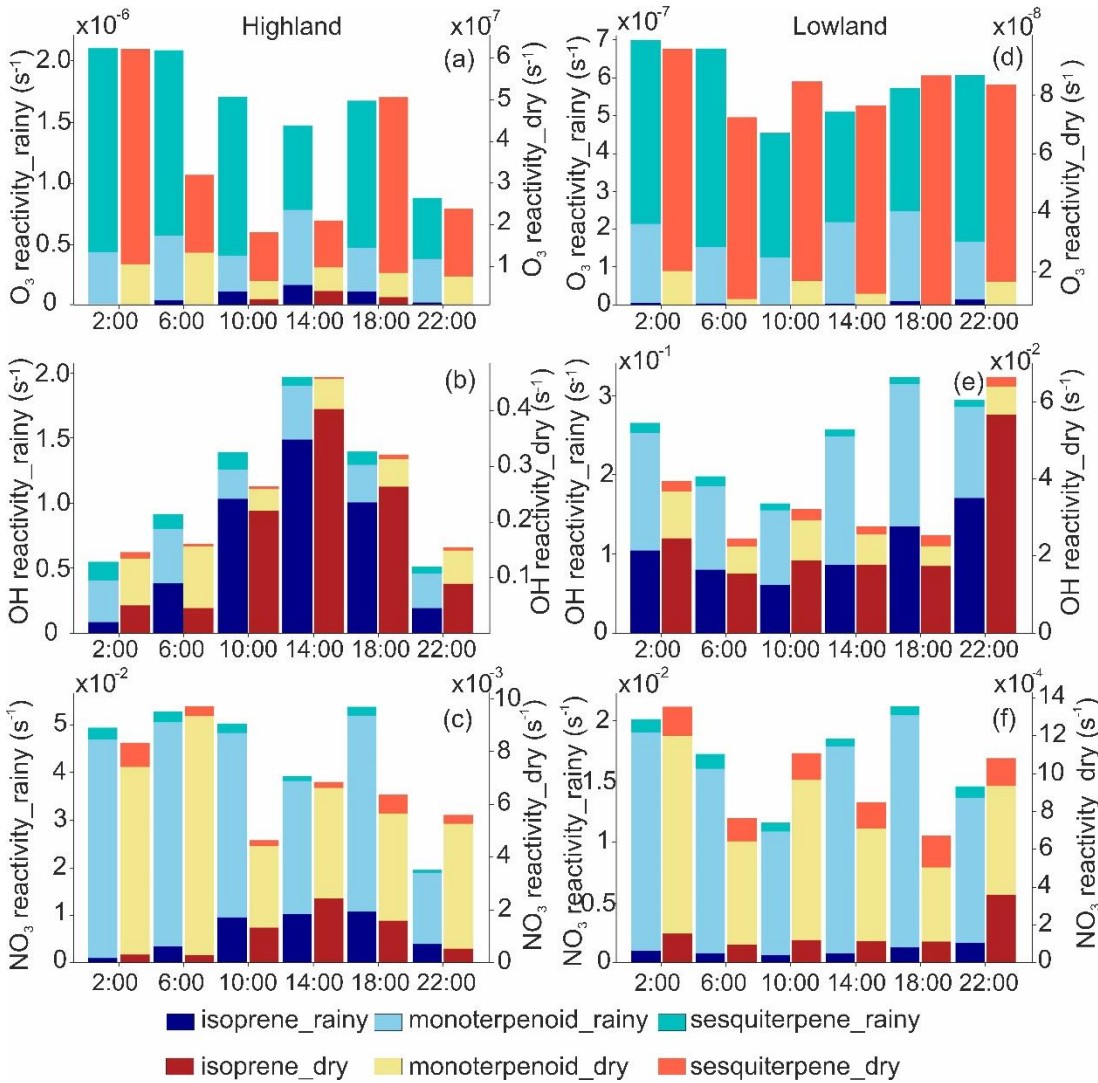

**Figure 6.** Reactivity of ozone (O₃), hydroxyl (OH), and nitrate (NO₃) of different biogenic volatile organic compounds in the highland **(a, b, c)** and lowland ecosystems **(d, e, f)** in the rainy and dry seasons.

## 3.3 Estimation of BVOC emission factors

The EFs for isoprene, MBO, and detected MTs, for the agriculture savanna ecosystem surrounding the Maktau site, were estimated for the rainy and dry seasons separately (Fig. 7). The median values of the EF for α-pinene, β-pinene, 3Δ-carene, camphene, and limonene (Figs. 7c to g) are higher during the rainy season in April than during the dry season in September, while the median values of the EF for MBO (Fig. 7b) and all other MTs (Fig. 7g) are higher during the dry season than during the rainy season. If the dependency of soil moisture availability on the emission of isoprene is considered, then the EF for isoprene during both the rainy and dry seasons is effectively the same (Fig. 7a). Considering the variability in the estimated EFs for the two different seasons, only the EFs for limonene show no overlap in the indicated error bars (Fig. 7f), which are defined by the minimum and maximum daily estimated EF. Thus, our results suggest that the EF for limonene might be seasonally dependent.

In order to put the estimated EFs into context and to contribute to an improved representation of BVOC emissions from African ecosystems in models, the estimated EFs are compared with the EFs used in MEGAN v2.1 for warm C4 grass and Crop1 (Guenther et al., 2012). The estimated EFs for isoprene (155 µg m⁻² h⁻¹ in the rainy season, 280 µg m⁻² h⁻¹ in the dry season) and β-pinene (2 µg m⁻² h⁻¹ in the rainy season, 1.5 µg m⁻² h⁻¹ in the dry season)

compare very well with the EFs used in MEGAN for warm C4 grass (Figs. 7a, d), and in the case of β-pinene, also for Crop1, since MEGAN assumes the same EF for β-pinene for the two different plant functional types. The estimated median EFs for MBO, α-pinene, 3Δ-carene, and limonene are higher than the EFs used in MEGANv2.1 by about 8 (4), 17 (53), 1 (2), and 89 (314) $\mu$g m$^{-2}$ h$^{-1}$, respectively, where the values in parenthesis are for the rainy season, while the others are for the dry season. The values of the estimated EFs compare best with the EFs allocated for warm C4 grass in MEGAN in the case of isoprene, α-pinene, β-pinene, and 3Δ-carene compared with the EFs for other plant functional types in MEGAN. However, the estimated EF for limonene is more in line with MEGAN's EF for tropical trees (80 $\mu$g m$^{-2}$ h$^{-1}$). Unfortunately, we could not identify the source of the limonene emitter(s). It could be the native African shrubs surrounding the lowland site, which are dominated by acacias (*Vachellia mellifera, VachelliaAcacia tortilis*), but to our knowledge emission rates have not been reported from these species. The EF for the sum of other MTs (i.e., six MTs at our site and up to 34 in MEGAN) is about 10 and 20 $\mu$g m$^{-2}$ h$^{-1}$ higher than that assumed in MEGAN for warm C4 grass and Crop1 for the rainy and dry seasons, respectively (Fig. 7g). During both seasons, linalool contributes the most to the total EF for the sum of other MTs in this study, while terpinolene accounts for the second largest fraction. Since the lifetime of monoterpenes is a few hours (see Sec. 3.2), it is likely that part of the detected monoterpenes have been transported to the site from areas covered by other plant functional types than warm C4 grass and Crop1, such as broadleaved trees and shrubs, which are thought to have a significantly higher potential to emit monoterpenes (Guenther et al., 2012). It is, however, noteworthy that our estimated EF for β-pinene is in line with the listed value by Guenther et al. (2012) for warm C4 grass and Crop1, but not for broadleaved trees and shrubs, though the lifetime of β-pinene is within the same range as that of the other monoterpenes. The estimated EF for MBO is much higher than that used for C4 grass in MEGAN. MBO has a lifetime of about half a day, and thus a great part of the detected MBO does not originate from the near vicinity of the site, but can have been transported far. However, the EF listed in Guenther et al. (2012) for MBO for all plant functional types present in the relevant parts of Africa (Ke et al., 2012) is still about 2-3 orders of magnitude lower than estimated here. This might call for a revision of EFs for MBO, considering that also Jaars et al. (2016) found even higher concentrations of MBO than we did in this study, in an area of Africa which also should not contain MBO emitting species.

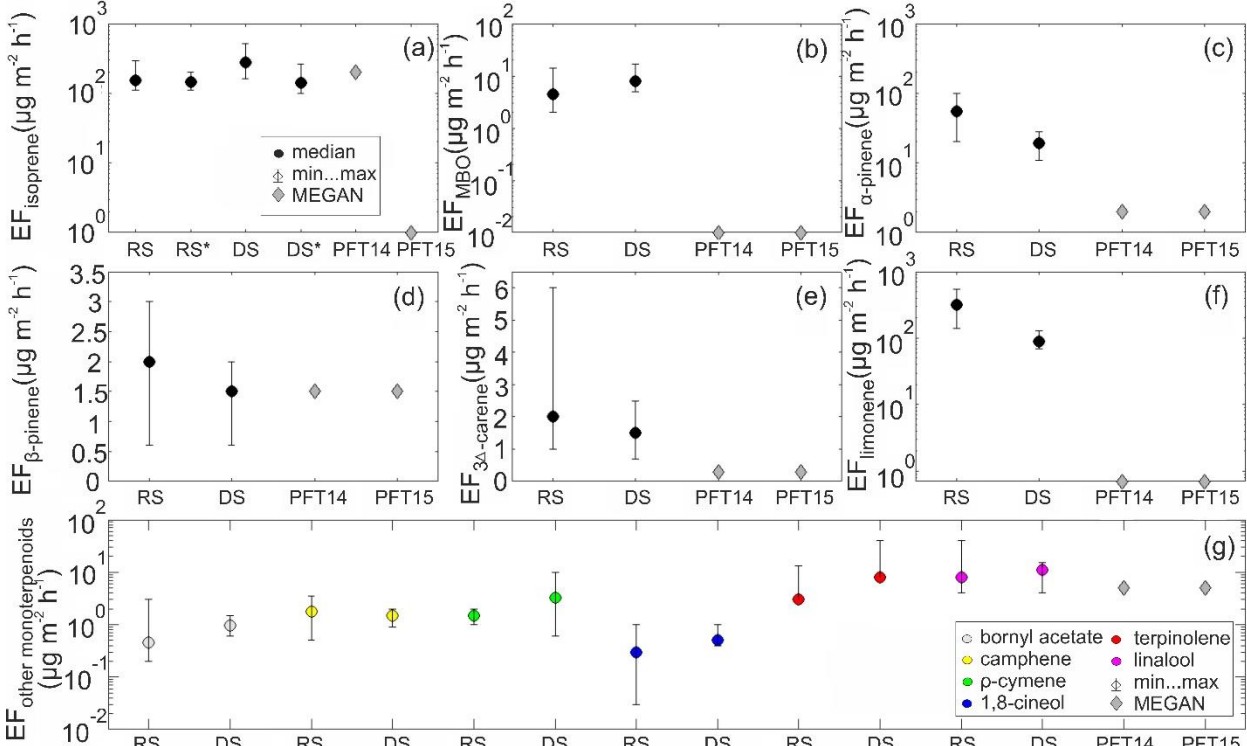

**Figure 7.** Estimated biogenic volatile organic compound (BVOC) emission factors (EF) for the agriculture savanna ecosystem surrounding the Maktau field site in comparison with the EFs for warm C4 grass (PFT14) and Crop1 (PFT15) used in MEGAN v2.1. The EFs have been estimated for the rainy season (RS, April) and for the dry season (DS, September) separately. **(a)** isoprene (The EF was estimated by either considering (RS, DS) or neglecting (RS*, DS*) a dependency of the activity factor on soil moisture availability), **(b)** 2-methyl-3-buten-2-ol, **(c)** α-pinene, **(d)** β-pinene, **(e)**

3Δ-carene, **(f)** limonene, and **(g)** other monoterpenes, which includes the sum of up to 34 other monoterpenes in MEGANv2.1 and includes bornyl acetate (gray), camphene (yellow), ρ-cymene (green), 1,8-cineol (blue), terpinolene (red), and linalool (magenta) at Maktau Weather Station. The legend provided in **(a)** is valid for **(a)** to **(f)**.

We emphasize that the estimated EFs are connected with a large degree of uncertainty, since they are not based on flux measurements from the site but are instead determined using observed BVOC mixing ratios and an inverse modeling

approach, which is limited by model assumptions and inputs.

## 4 Conclusion

In this study we measured mixing ratios of isoprene, MTs, SQTs, bornyl acetate, and MBO in the humid highland and dry lowland ecosystems in Taita Taveta County, southern Kenya, during both a rainy and a dry season.

Isoprene and MTs showed the highest mixing ratios in both the highlands and lowlands, while α-pinene, limonene,

and β-pinene accounted for the largest contribution to the total mixing ratio of MTs. Isoprene dominated the total BVOC mixing ratio during daytime and reached diurnal peak mixing ratios in the afternoon in the highlands and in the early evening in the lowlands. The mixing ratio of MTs generally peaked between midnight and early morning, and MTs dominated the total BVOC mixing ratio during nighttime. Isoprene was the dominant BVOC contributor to the OH reactivity, MTs dominated the $NO_3$ reactivity of BVOCs, and SQTs showed higher contributions to the $O_3$ reactivity of

BVOCs than isoprene and MTs.

Using an inverse model approach with measured BVOC mixing ratios and meteorology data, we estimated the EFs for isoprene, MBO, and MTs in the agriculture savanna ecosystem. The estimated EFs for isoprene and β-pinene agreed very well with what is currently assumed in MEGANv2.1, for warm C4 grass, but the estimated EFs for MBO, α-pinene, and especially limonene were significantly higher than what is assumed in MEGAN for the relevant plant functional type.
Additionally, our results indicate that the EF for limonene might be seasonally dependent.

**Appendix A:**

**Table A1**. Sample and flow rate measurements in Wundanyi and Maktau field site during the rainy and dry season (sccm: standard cubic centimeters per minute).

|  | Wundanyi Station (humid highland) | | Maktau field site (dry lowland) | |
| --- | --- | --- | --- | --- |
|  | First campaign | Second campaign | First campaign | Second campaign |
| Rainy season (2019) | Apr 10th to 13th<br>4 h/tube<br>Total: 22 tubes<br>100.5 sccm | Apr 13th to 15th<br>2 h/tube<br>Total: 24 tubes<br>99.5 sccm | Apr 10th to 13th<br>4 h/tube<br>Total: 22 tubes<br>70 sccm | Apr 14th to 17th<br>2 h/tube<br>Total: 24 tubes<br>67 sccm |
| Dry season (2019) | Sept 1st to 5th<br>4h/tube<br>Total: 24 tubes<br>85.5 sccm | Sept 10th to 14th<br>4h/tube<br>Total: 24 tubes<br>81.5 sccm | Sept 6th to 10th<br>4h/tube<br>Total: 24 tubes<br>83 sccm | Sept 16th to 19th<br>4h/tube<br>Total: 24 tubes<br>92 sccm |


**Table A2.** The analytical uncertainty (U) for ambient temperature and the limit-of-quantification (LOQ) for the studied compounds (The estimation method is described in Helin et al. (2020)).

| Compounds | U (%) | LOQ (pptv) |
|---|---|---|
| Isoprene | 20 | 2.5 |
| α-Pinene | 17 | 0.5 |
| Camphene | 17 | 0.1 |
| β-Pinene | 17 | 0.2 |
| Limonene | 17 | 0.9 |
| ρ-Cymene | 17 | 0.3 |
| 3Δ-Carene | 16 | 0.3 |
| 18-Cineol | 18 | 0.4 |
| Terpinolene | 19 | 1.0 |
| Linalool | 20 | 1.2 |
| Longicyclene | 19 | 0.3 |
| Iso-longifolene | 20 | 0.4 |
| β-Caryophyllene | 18 | 1.0 |
| β-Farnesene | 23 | 2.0 |
| α-Humulene | 19 | 0.3 |
| MBO | 20 | 0.9 |
| Bornyl acetate | 21 | 0.5 |

**Table A3.** Reaction rate coefficients ($k_{i,x}$) applied in the model used for estimations of the emission factors and for reactivity calculations. T (K) is air temperature.

| Compound | $k_{O3}$ (cm³ s⁻¹) | Reference | $k_{OH}$ (cm³ s⁻¹) | Reference |
|---|---|---|---|---|
| Isoprene | $1.03 \cdot 10^{-14}$ $\cdot e\left(\frac{-1995}{T}\right)$ | IUPAC preferred value (http://iupac.pole-ether.fr/htdocs/datasheets/xhtml/Ox_VOC7_O3_CH2C(CH3)CHCH2.xhtml_mathml.xml) | $2.70 \cdot 10^{-11}$ $\cdot e\left(\frac{390}{T}\right)$ | Master Chemical Mechanism, MCM v3.2 (Jenkin et al., 1997; Saunders et al., 2003), via website: http://mcm.leeds.ac.uk/MCM |
| MBO | $1.0 \cdot 10^{-17}$ | Grosjean and Grosjean (1994) | $8.1 \cdot 10^{-12}$ $\cdot e\left(\frac{610}{T}\right)$ | Rudich et al. (1995) |
| Bornyl acetate | - | bornyl acetate does not react with O₃, because it is a saturated hydrocarbon | $13.9 \cdot 10^{-12}$ | Coeur et al. (1999) |
| $\alpha$-Pinene | $8.05 \cdot 10^{-16}$ $\cdot e\left(\frac{-640}{T}\right)$ | IUPAC preferred value (http://iupac.pole-ether.fr/htdocs/datasheets/pdf/Ox_VOC8_O3_apinene.pdf) | $1.2 \cdot 10^{-11}$ $\cdot e\left(\frac{440}{T}\right)$ | IUPAC preferred value (http://iupac.pole-ether.fr/htdocs/datasheets/xhtml/HOx_VOC9_HO_apinene.xhtml_mathml.xml) |
| Camphene | $9.0 \cdot 10^{-19}$ | Atkinson (1997) | $5.3 \cdot 10^{-11}$ | Atkinson (1997) |
| $\beta$-Pinene | $1.35 \cdot 10^{-15}$ $\cdot e\left(\frac{-1270}{T}\right)$ | IUPAC preferred value (http://iupac.pole-ether.fr/htdocs/datasheets/pdf/Ox_VOC19_O3_bpinene.pdf) | $2.38 \cdot 10^{-11}$ $\cdot e\left(\frac{357}{T}\right)$ | Kleindienst et al. (1982) |
| 3Δ-Carene | $3.7 \cdot 10^{-17}$ | Atkinson (1997) | $8.8 \cdot 10^{-11}$ | Atkinson (1997) |
| ρ-Cymene | $5.0 \cdot 10^{-20}$ | Hellén et al. (2018) | $1.5 \cdot 10^{-11}$ | Corchnoy and Atkinson (1990) |
| Limonene | $2.80 \cdot 10^{-15}$ $\cdot e\left(\frac{-770}{T}\right)$ | IUPAC preferred value (http://iupac.pole-ether.fr/htdocs/datasheets/pdf/Ox_VOC20_O3_limonene.pdf) | $4.28 \cdot 10^{-11}$ $\cdot e\left(\frac{401}{T}\right)$ | Gill and Hites (2002) |
| 1,8-Cineol | $1.5 \cdot 10^{-19}$ | Hellén et al. (2018) | $1.11 \cdot 10^{-11}$ | Corchnoy and Atkinson (1990) |
| Terpinolene | $1.88 \cdot 10^{-15}$ | Shu and Atkinson (1994) | $22.5 \cdot 10^{-11}$ | Corchnoy and Atkinson (1990) |
| Linalool | $4.3 \cdot 10^{-16}$ | Atkinson et al. (1995) | $15.9 \cdot 10^{-11}$ | Atkinson et al. (1995) |

**Table A4**. Data variation and $O_3$, OH and $NO_3$ reaction rate coefficients ($k_{i,x}$, values were shown in daily average) for various BVOCs (RSD: relative standard deviation, unit in %; k unit in $cm^3s^{-1}$; $\tau$ unit in hour)

| Compound | Season | RSD | k_O₃ | k_OH | k_NO₃ | τ _O₃ | τ_OH |
|---|---|---|---|---|---|---|---|
| Isoprene | Maktau_rainy | 101.19 | 1.26E-17 | 1.00E-10 | 6.50E-13 | 32.6 | 2.2 |
| | Maktau_dry | 80.34 | 1.18E-17 | 1.02E-10 | 6.39E-13 | 33.5 | 2.4 |
| | Wundanyi_rainy | 112.83 | 1.15E-17 | 1.02E-10 | 6.37E-13 | 31.9 | 2.2 |
| | Wundanyi_dry | 112.24 | 1.07E-17 | 1.04E-10 | 6.26E-13 | 37.1 | 2.4 |
| α-Pinene | Maktau_rainy | 96.35 | 9.36E-17 | 5.27E-11 | 6.23E-12 | 4.4 | 4.2 |
| | Maktau_dry | 117.30 | 9.15E-17 | 5.35E-11 | 6.35E-12 | 4.3 | 4.6 |
| | Wundanyi_rainy | 48.39 | 9.09E-17 | 5.38E-11 | 6.38E-12 | 4.0 | 4.1 |
| | Wundanyi_dry | 124.54 | 8.88E-17 | 5.46E-11 | 6.49E-12 | 4.5 | 4.5 |
| Camphene | Maktau_rainy | 111.89 | 9.0E-19 | 5.3E-11 | 6.6E-13 | 458.6.9 | 4.2 |
| | Maktau_dry | 272.87 | 9.0E-19 | 5.3E-11 | 6.6E-13 | 437.9 | |
| | Wundanyi _rainy | 194.68 | 9.0E-19 | 5.3E-11 | 6.6E-13 | 408.6 | |
| | Wundanyi dry | 90.17 | 9.0E-19 | 5.3E-11 | 6.6E-13 | 440.8 | |
| β-Pinene | Maktau_rainy | 146.87 | 1.89E-16 | 7.90.E-10 | 2.5E-12 | 21.8 | 2.8 |
| | Maktau_dry | 189.67 | 1.81E-17 | 8.01E-11 | 2.5E-12 | 21.8 | 3.1 |
| | Wundanyi _rainy | 74.68 | 1.78E-17 | 8.04E-11 | 2.5E-12 | 20.6 | 2.8 |
| | Wundanyi _dry | 102.85 | 1.70E-17 | 8.14E-11 | 2.5E-12 | 23.3 | 3.0 |
| Limonene | Maktau_rainy | 93.67 | 2.10E-17 | 1.65E-11 | 1.2E-11 | 2.0 | 1.3 |
| | Maktau_dry | 67.97 | 2.05E-16 | 1.67E-10 | 1.2E-11 | 1.9 | 1.5 |
| | Wundanyi _rainy | 53.66 | 2.03E-16 | 1.68E-10 | 1.2E-11 | 1.8 | 1.3 |
| | Wundanyi _dry | 142.89 | 1.97E-16 | 1.70E-10 | 1.2E-11 | 2.0 | 1.4 |
| ρ-Cymene* | Maktau_rainy | 74.49 | 5.00E-20 | 1.11E-11 | | 8254.9 | 14.7 |
| | Maktau_dry | 156.80 | 5.00E-20 | 1.11E-11 | | 7882.7 | 16.2 |
| | Wundanyi_rainy | 107.31 | 5.00E-20 | 1.11E-11 | | 7354.1 | 14.7 |
| | Wundanyi_dry | 96.52 | 5.00E-20 | 1.11E-11 | | 7934.0 | 16.2 |
| 3Δ-Carene | Maktau_rainy | 129.99 | 3.7E-17 | 2.25E-11 | 9.1E-12 | 11.2 | 2.5 |
| | Maktau_dry | 146.98 | 3.7E-17 | 2.25E-11 | 9.1E-12 | 10.7 | 2.8 |
| | Wundanyi_rainy | 79.57 | 3.7E-17 | 2.25E-11 | 9.1E-12 | 9.9 | 2.5 |
| | Wundanyi_dry | 107.19 | 3.7E-17 | 2.25E-11 | 9.1E-12 | 10.7 | 2.8 |
| 18-Cineol | Maktau_rainy | 195.09 | 1.5E-19 | 1.59E-11 | | 2751.6 | 20.0 |

| | | | | | | | |
|---|---|---|---|---|---|---|---|
| | Maktau_dry | 88.31 | 1.5E-19 | 1.59E-11 | | 2627.6 | 22.1 |
| | Wundanyi_rainy | 107.14 | 1.5E-19 | 1.59E-11 | | 2451.4 | 20.0 |
| | Wundanyi_dry | 214.96 | 1.5E-19 | 1.59E-11 | | 2644.7 | 22.1 |
| Terpinolene | Maktau_rainy | 141.35 | 1.88E-15 | 2.25E-10 | 9.7E-11 | 0.2 | 1.0 |
| | Maktau_dry | 284.61 | 1.88E-15 | 2.25E-10 | 9.7E-11 | 0.2 | 1.1 |
| | Wundanyi_rainy | 84.71 | 1.88E-15 | 2.25E-10 | 9.7E-11 | 0.2 | 1.0 |
| | Wundanyi_dry | 133.40 | 1.88E-15 | 2.25E-10 | 9.7E-11 | 0.2 | 1.1 |
| Linalool | Maktau_rainy | 141.69 | 4.3E-16 | 1.59E-10 | 1.10E-11 | 1.0 | 1.4 |
| | Maktau_dry | 73.08 | 4.3E-16 | 1.59E-10 | 1.10E-11 | 0.9 | 1.5 |
| | Wundanyi_rainy | 467.26 | 4.3E-16 | 1.59E-10 | 1.10E-11 | 0.9 | 1.4 |
| | Wundanyi_dry | 120.62 | 4.3E-16 | 1.59E-10 | 1.10E-11 | 0.9 | 1.5 |
| Longicyclene | Maktau_rainy | 96.75 | 5.0E-19 | 4.7E-11 | 6.8E-13 | 825.5 | 4.7 |
| | Maktau_dry | 618.05 | 5.0E-19 | 4.7E-11 | 6.8E-13 | 788.3 | 5.2 |
| | Wundanyi_rainy | 110.58 | 5.0E-19 | 4.7E-11 | 6.8E-13 | 735.4 | 4.7 |
| | Wundanyi_dry | 133.07 | 5.0E-19 | 4.7E-11 | 6.8E-13 | 793.4 | 5.2 |
| Iso-longifolene | Maktau_rainy | 326.40 | 1.0E-17 | | 3.9E-12 | 41.3 | |
| | Maktau_dry | 618.05 | 1.0E-17 | | 3.9E-12 | 39.4 | |
| | Wundanyi_rainy | 366.74 | 1.0E-17 | | 3.9E-12 | 36.8 | |
| | Wundanyi_dry | 685.57 | 1.0E-17 | | 3.9E-12 | 39.7 | |
| β-Caryophyllene | Maktau_rainy | 89.74 | 1.2E-14 | 2.0E-10 | 1.9E-11 | 0.0 | 1.1 |
| | Maktau_dry | 93.96 | 1.2E-14 | 2.0E-10 | 1.9E-11 | 0.0 | 1.2 |
| | Wundanyi_rainy | 100.02 | 1.2E-14 | 2.0E-10 | 1.9E-11 | 0.0 | 1.1 |
| | Wundanyi_dry | 276.94 | 1.2E-14 | 2.0E-10 | 1.9E-11 | 0.0 | 1.2 |
| β-Farnesene | Maktau_rainy | 134.45 | 5.59E-16 | 2.3E-10 | 1.9E-11 | 0.7 | 1.0 |
| | Maktau_dry | 76.79 | 5.59E-16 | 2.3E-10 | | 0.7 | 1.1 |
| | Wundanyi_rainy | 144.76 | 5.59E-16 | 2.3E-10 | | 0.7 | 1.0 |
| | Wundanyi_dry | 113.73 | 5.59E-16 | 2.3E-10 | | 0.7 | 1.1 |
| α-Humulene | Maktau_rainy | 61.93 | 1.2E-14 | 2.9E-10 | 3.5E-11 | 0.0 | 0.8 |
| | Maktau_dry | 60.92 | 1.2E-14 | 2.9E-10 | 3.5E-11 | 0.0 | 0.8 |
| | Wundanyi_rainy | 78.56 | 1.2E-14 | 2.9E-10 | 3.5E-11 | 0.0 | 0.8 |
| | Wundanyi_dry | 87.74 | 1.2E-14 | 2.9E-10 | 3.5E-11 | 0.0 | 0.8 |
| MBO | Maktau_rainy | 69.37 | 1.0E-17 | 6.30E-11 | 1.20E-14 | 41.3 | 3.5 |

|  |  |  |  |  |  |  |  |
|---|---|---|---|---|---|---|---|
|  | Maktau_dry | 68.78 | 1.0E-17 | 6.44E-11 | 1.20E-14 | 39.4 | 3.8 |
|  | Wundanyi_rainy | 92.89 | 1.0E-17 | 6.48E-11 | 1.20E-14 | 36.8 | 3.4 |
|  | Wundanyi_dry | 109.41 | 1.0E-17 | 6.63E-11 | 1.20E-14 | 39.7 | 3.7 |
| Bornyl acetate | Maktau_rainy | 174.73 |  | 1.39E-11 |  |  | 16.0 |
|  | Maktau_dry | 77.48 |  | 1.39E-11 |  |  | 17.6 |
|  | Wundanyi_rainy | 78.44 |  | 1.39E-11 |  |  | 16.0 |
|  | Wundanyi_dry | 77.05 |  | 1.39E-11 |  |  | 17.6 |

Calculation methods of k values were shown in Table A3.

**Table A5.** Minimum sum of the squared differences (MSSD) between the predicted and observed VOC mixing ratios for each campaign day presented alongside the emission factor (EF), for each VOC, which is calculated to be the most 485 appropriate value for each day. $MSSD_i = \sum_{j=1}^{n}([BVOC_i]_{measurment,j} - [BVOC_i]_{model,j})^2$, where n is the number of measured VOC mixing ratios samples and i is the 12 different VOCs for which the mixing ratios were predicted.

| | | MSSD (ppt$^2$) / EF$_{day}$ ($\mu$g m$^{-2}$ h$^{-1}$) | | | | | |
|---|---|---|---|---|---|---|---|
| Date | n | Isoprene | MBO | α-Pinene | $\beta$-Pinene | 3-Carene | Limonene |
| 11.04.2019 | 3 | 309/290 | 9/14 | 181/60 | 6/3 | 1/2 | 2985/500 |
| 12.04.2019 | 3 | 1591/130 | 5/4 | 329/60 | 0.4/2 | 0.1/1 | 148/140 |
| 13.04.2019 | 3 | 54/140 | 9/3 | 51/30 | 0.7/3 | 0.4/2 | 51/300 |
| 14.04.2019 | 2 | 14/110 | 3/2 | 2/20 | 0.05/0.6 | 0.06/1 | 6/240 |
| 15.04.2019 | 3 | 2221/200 | 0.7/5 | 1246/100 | 0.9/2 | 0.7/2 | 272/330 |
| 16.04.2019 | 5 | 2502/170 | 19/9 | 550/50 | 4/2 | 30/6 | 12630/540 |
| 06.09.2019 | 3 | 17/270 | 0.03/9 | 0.2/20 | 0.01/1.5 | 0.008/2 | 0.1/70 |
| 07.09.2019 | 3 | 30/340 | 0.02/8 | 0.06/11 | 0.002/1 | 0.0009/2 | 3/100 |
| 08.09.2019 | 3 | 61/520 | 0.9/17 | 0.1/18 | 0.01/1.5 | 0.002/1.5 | 0.4/70 |
| 09.09.2019 | 3 | 0.7/290 | 0.001/8 | 0.5/28 | 0.004/2 | 0.003/2.5 | 0.2/80 |
| 15.09.2019 | 3 | 206/160 | 1/7 | 0.9/28 | 0.004/2 | 0.005/1.5 | 6/120 |
| 16.09.2019 | 3 | 52/450 | 1/8 | 0.2/28 | 0.0008/1.5 | 0.003/0.9 | 0.09/70 |
| 17.09.2019 | 3 | 41/210 | 0.05/5 | 0.3/15 | 0.006/1.5 | 0.002/0.7 | 5/130 |
| 18.09.2019 | 3 | 278/210 | 0.4/8 | 0.2/17 | 0.007/0.6 | 0.007/1.5 | 3/110 |

Continued  Table A5

| | | MSSD (ppt$^2$) / EF$_{day}$ ($\mu$g m$^{-2}$ h$^{-1}$) | | | | | |
|---|---|---|---|---|---|---|---|
| Date | n | Bornyl acetate | Camphene | ρ-Cymene | 1,8-Cineol | Terpinolene | Linalool |
| 11.04.2019 | 3 | 32/3 | 3/3.5 | 0.7/2 | 3/1 | 0.009/2 | 16/40 |
| 12.04.2019 | 3 | 0.04/0.4 | 2/2 | 14/2 | 3/0.6 | 0.03/4 | 0.2/7 |
| 13.04.2019 | 3 | 0.06/0.5 | 0.2/1 | 0.3/1 | 0.3/0.2 | 0.01/2 | 0.1/6 |
| 14.04.2019 | 2 | 0.002/0.2 | 0.02/0.5 | 0.7/1 | 0.00006/0.08 | 0/0 | 0.01/4 |
| 15.04.2019 | 3 | 0.1/0.4 | 1/1.5 | 8/2 | 0.01/0.03 | 0.02/13 | 0.1/15 |
| 16.04.2019 | 5 | 0.7/1 | 4/2.5 | 1/1 | 1/0.4 | 0.06/10 | 0.2/9 |
| 06.09.2019 | 3 | 0.001/1.5 | 0.003/1.5 | 19/10 | 0.006/0.8 | 0/0 | 0.004/4 |
| 07.09.2019 | 3 | 0.007/1 | 0.005/1.5 | 0.06/3 | 0.0004/0.5 | 0.004/13 | 0.004/14 |
| 08.09.2019 | 3 | 0.006/0.9 | 0.006/1.5 | 0.07/4 | 0.0003/0.5 | 0/0 | 0.0003/12 |
| 09.09.2019 | 3 | 0.01/1.5 | 0.02/1.5 | 0.4/3.5 | 0.0006/0.5 | 0.007/8 | 0.006/6 |
| 15.09.2019 | 3 | 0.0006/1 | 0.002/1 | 0.02/1.5 | 0.1/1 | 0.005/8 | 0.02/15 |
| 16.09.2019 | 3 | 0.003/0.6 | 0.008/0.9 | 0.003/0.6 | 0.01/0.4 | 0.005/20 | 0.0007/10 |
| 17.09.2019 | 3 | 0.0003/0.6 | 0.003/1.5 | 9/7 | 0.03/0.6 | 0.04/40 | 0.007/12 |
| 18.09.2019 | 3 | 0.004/0.7 | 0.04/2 | 0.01/1.5 | 0.0008/0.5 | 0.005/6 | 0.006/9 |

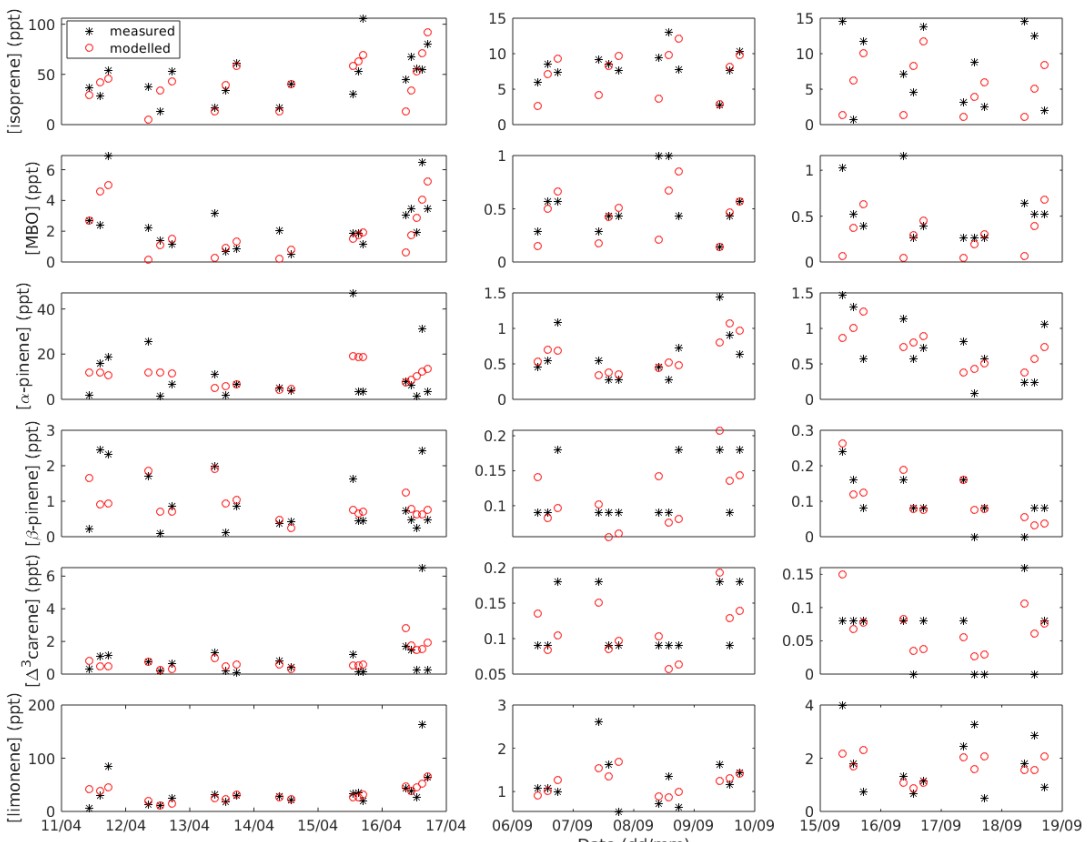

**Figure A1−1.** Measured and modelled mixing ratios of BVOCs.


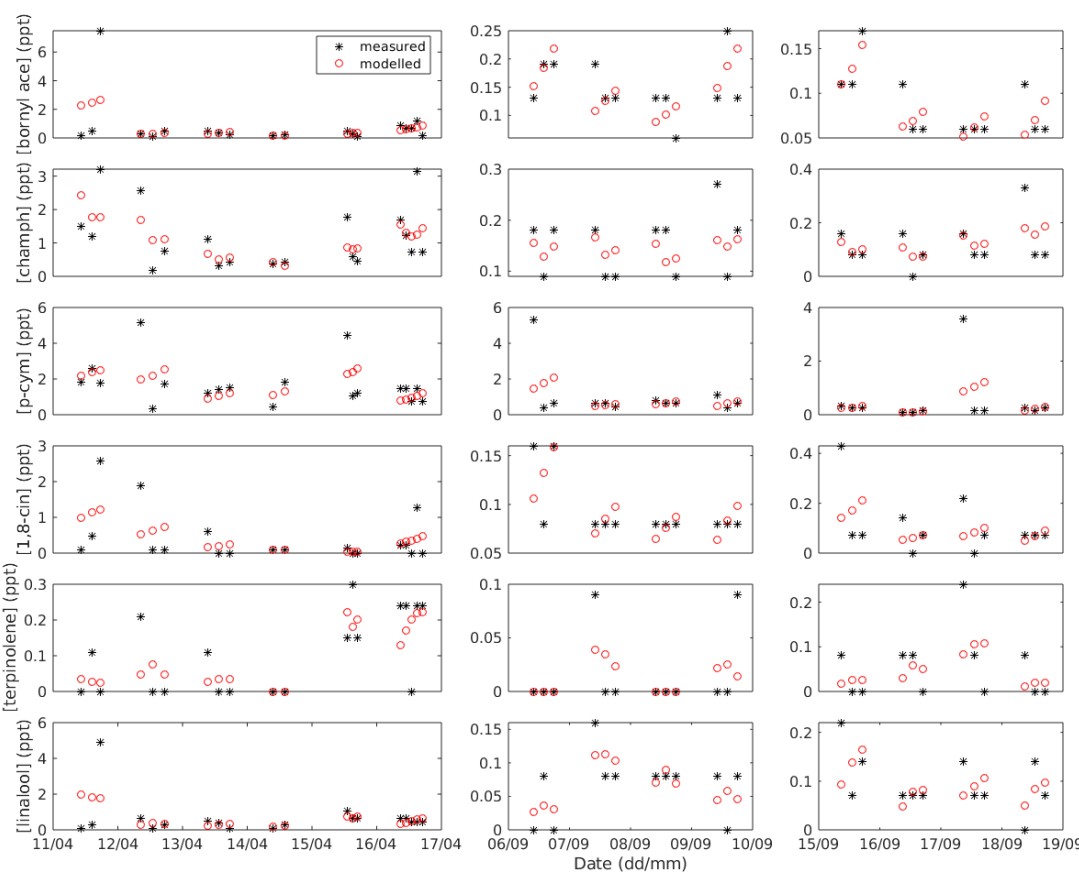

**Figure A1−2.** Measured and modelled mixing ratios of BVOCs.

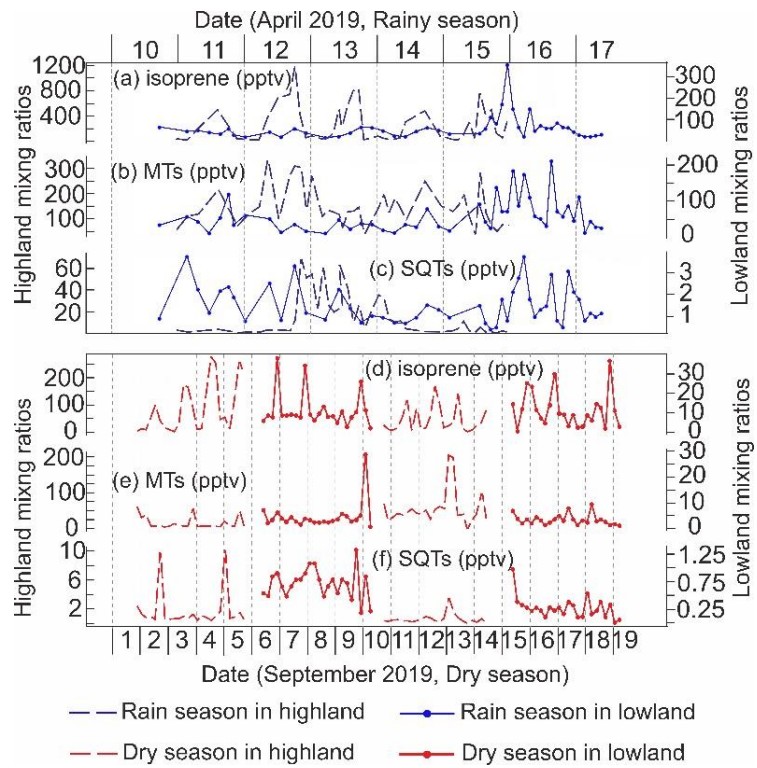

**Figure A2.** The temporal variability of biogenic volatile organic compound mixing ratios in the highland and lowland ecosystems during the rainy and dry seasons: **(a, d)** isoprene, **(b, e)** monoterpenoids (MTs), **(c, f)** sesquiterpenes (SQTs).

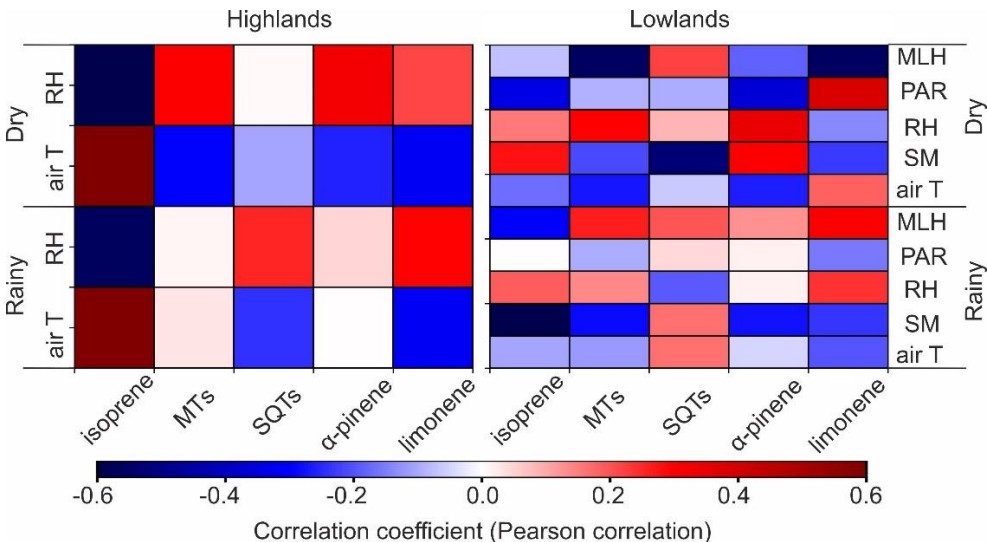

**Figure A3**. Correlation coefficients between BVOCs and environmental factors.

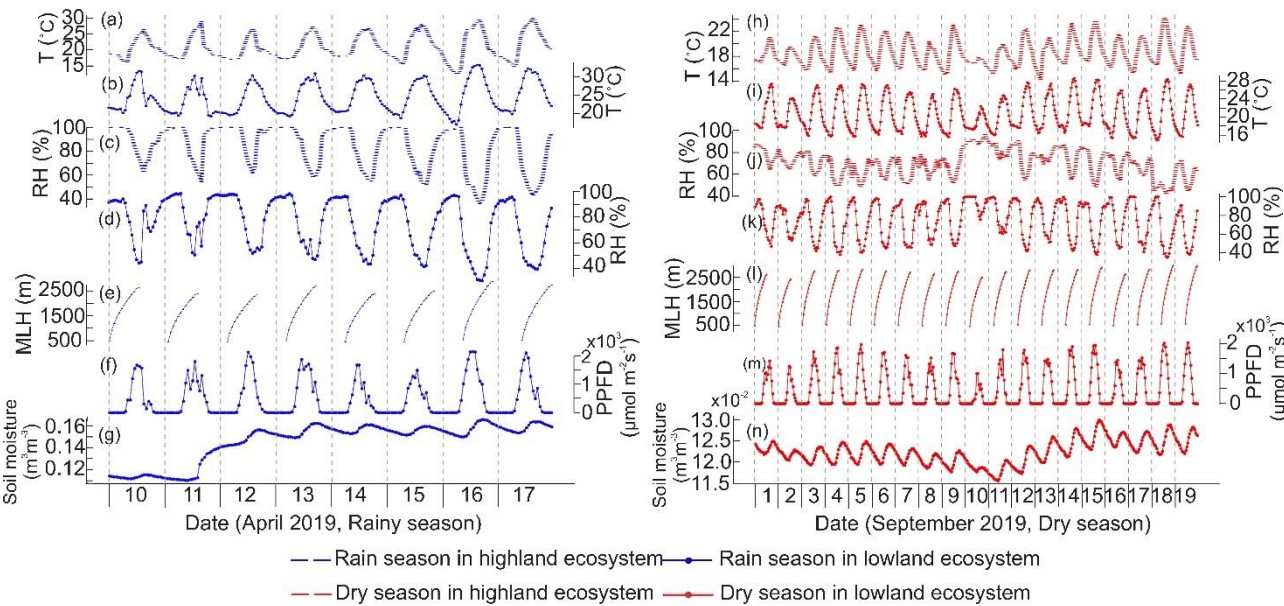

**Figure A4.** Meteorological measurements in the highland and lowland ecosystems during the rainy and dry seasons. **(a, h)** air temperature (T) in the highlands, **(b, i)** T in the lowlands, **(c, j)** relative humidity (RH) in the highlands, **(d, k)** RH in the lowlands, **(e, l)** mixing layer height (MLH) in the lowlands, **(f, m)** photosynthetic photon flux density (PPFD) in the lowlands, **(g, n)** soil moisture in the lowlands.

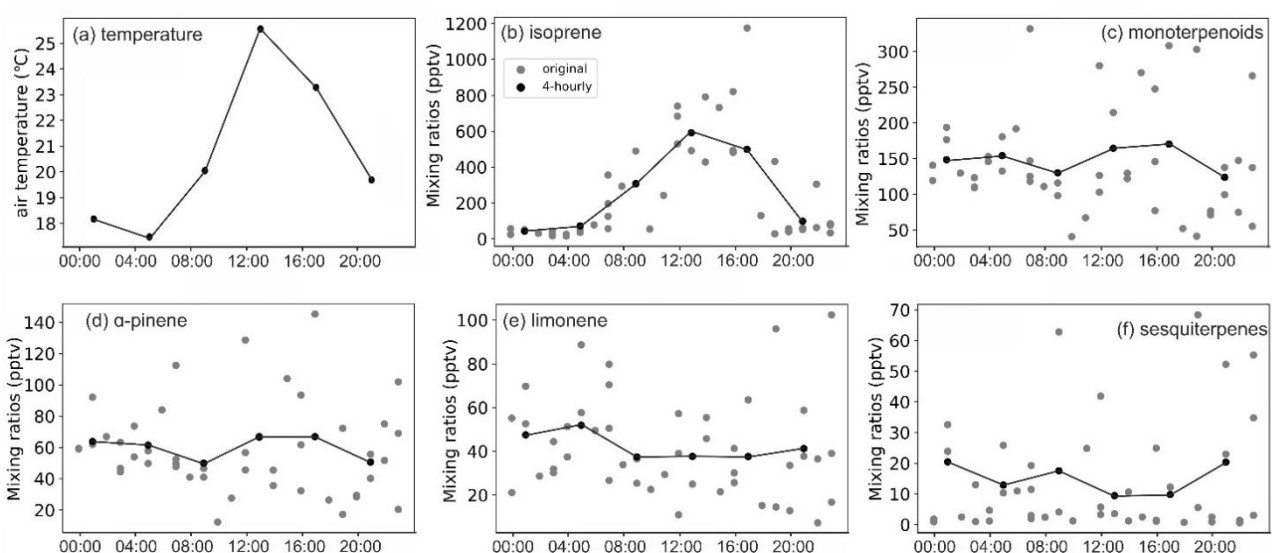

**Figure A5−1**. BVOC mixing ratios in the highland site during the rainy season. Fig. A5−1 (b) to (f) shared the same labels.

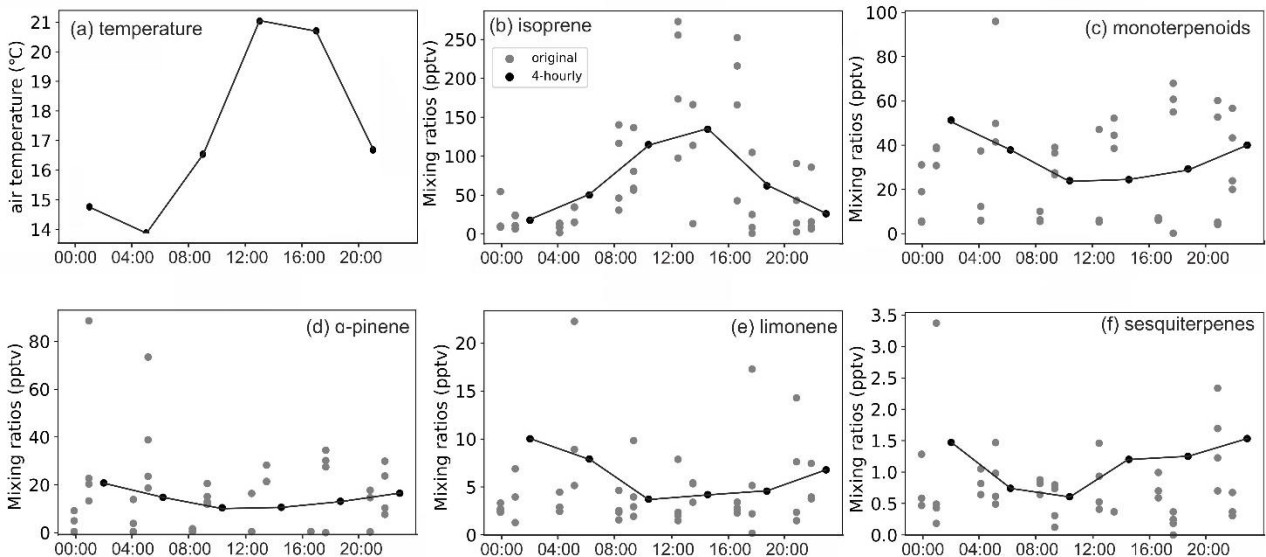

**Figure A5−2.** BVOC mixing ratios in the highland site during the dry season. Fig. A5−2 (b) to (f) shared the same labels.

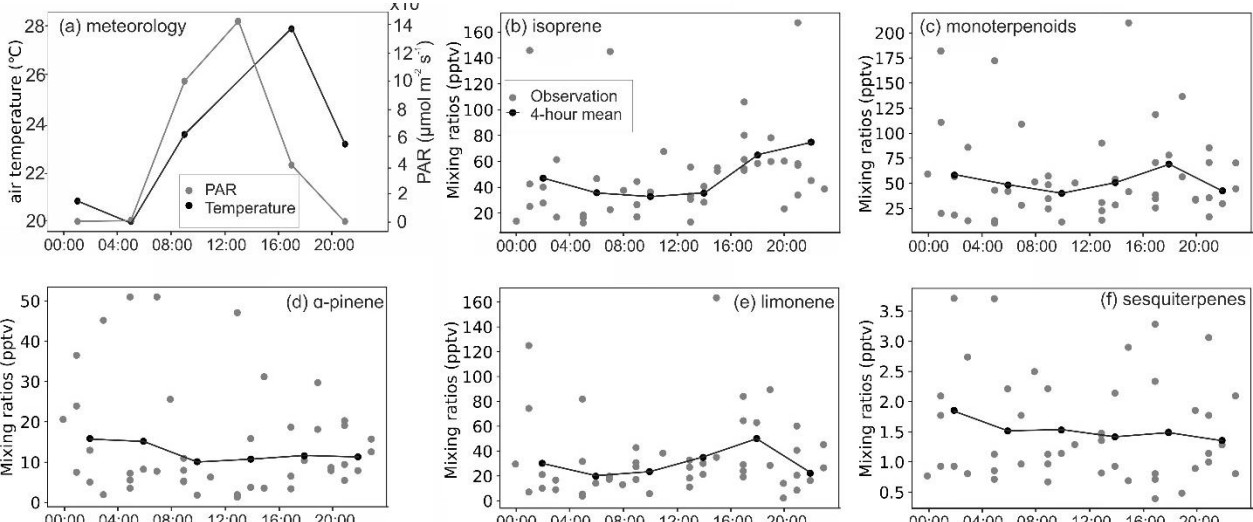

**Figure A5−3**. BVOC mixing ratios in the lowland site during the rainy season. Fig. A5−3 (b) to (f) shared the same labels.

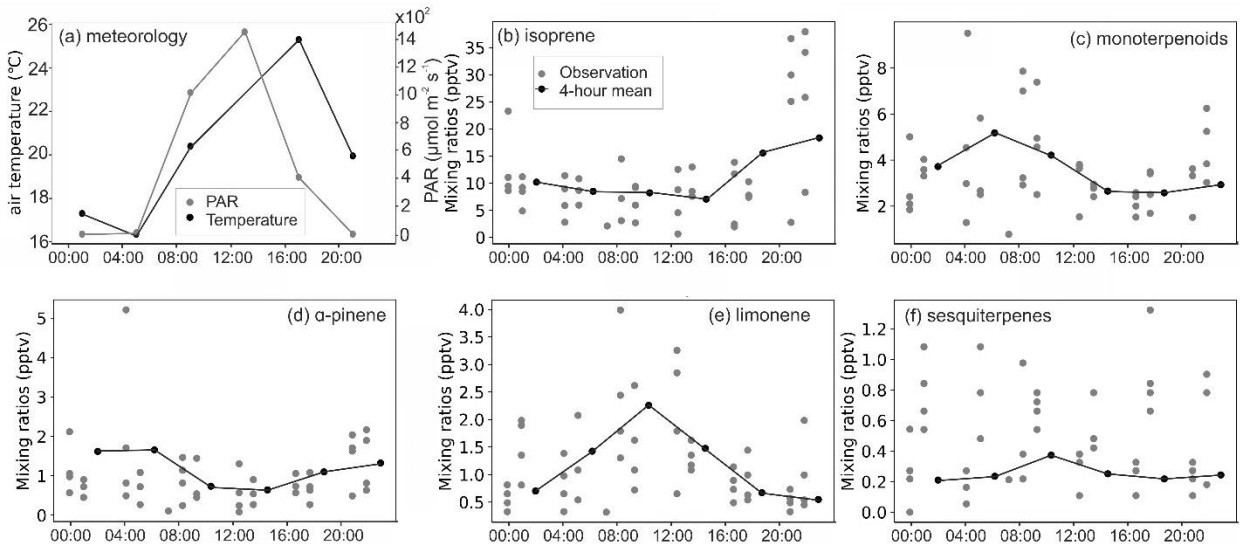

**Figure A5−4**. BVOC mixing ratios in the lowland site during the dry season. Fig. A5−4 (b) to (f) shared the same labels.

*Data availability.* BVOC mixing ratios and meteorological data used in this work are available from the authors upon request *(yang.z.liu@helsinki.fi and petri.pellikka@helsinki.fi).*

*Author contributions.* Y.L., S.S., H.H., and P.P. planned the measurement protocol and Y.L., S.S., L.M., and P.P. performed the measurements in Kenya, while T.T. conducted the laboratory analysis in Finland. Y.L., T.T., and H.H. performed the data interpretation and analysis. M.R. calculated MLHs. D.T. developed the BVOC emissions and 520 chemistry model for estimating BVOC EFs, conducted the simulations, and wrote the sections related to this work. Y.L. wrote the paper with contributions from all authors. The final version of the manuscript was approved by all authors.

*Competing interests.* The authors declare that they have no conflict of interest.

*Acknowledgements.* This work was supported by the University of Helsinki and its Taita Research Station, the Finnish Meteorology Institute, the Mazingira Centre of the International Livestock Research Institute, the China Scholarship 525 Council fellowship (funding no. 201806040217), Academy of Finland projects (nos 318645, 316151, 323255, 307957, and 275608), and Academy of Finland Flagship funding (grant no. 337552). Research permit P/18/97336/26355 from the National Council for Science and Technology of Kenya is greatly acknowledged. We appreciate Ms. Cathryn Primrose-Mathisen for language editing, the staff of the Taita Research Station of the University of Helsinki for the logistics and Mr. Mjomba Mwadime for his help with sample collection.

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
