# Peer review of "Seasonal and diurnal variations of biogenic volatile organic compounds in highland and lowland ecosystems in southern Kenya"

_Atmospheric Chemistry and Physics, 2021_

## Author Comment (AC1)

Thank you for the valuable comments and corrections! We have considered them carefully and have added more information related to the sample collection, chemical analysis and the inverse model in our manuscript, the tables and figures were also revised in accordance with your and Reviewer #1's suggestions. Our replies were addressed below.

**Anonymous Reviewer #2**

*The study by Liu et al. reports on ambient observations of BVOCs in southern Kenya and investigates the differences in their mixing ratios at contrasting environments (highlands vs lowlands). In addition to the comprehensive presentation of their observations, the authors attempt to characterize the significance of local BVOCs in atmospheric chemistry, through calculations of the OH, O3 and NO3 reactivities, and eventually, they use an inverse modelling approach to calculate and compare the emission factors with the MEGAN model. This is a well written study that presents observations from an environment that such data are scarce and therefore valuable. I recommend publication of the study after the authors address the comments from the excellent review of Anonymous reviewer1 and the following minor additional points.*

*2.2 How many samples were collected per season and per site?*

Thanks for your clarifying question. We have now added the information to Appendix A (Table A1).

*L181. The provided link does not work.*

Sorry about that. We have now added another link (Line 193).

*L267. Figure 5 is quite confusing. Maybe it would be better if the results are plotted in different figures for the wet and the dry seasons. In any case, the color selection has space for improvement. The authors may want to use a tool for selecting appropriate colors (e.g. https://colorbrewer2.org ).*

Thanks for your suggestions and link. We tried to incorporate both your and reviewer #1's suggestions regarding Fig. 5 (thus see reply to Ref #1 regarding comments on Fig. 5). In practice Fig. 5 was modified and additional figures displaying the daily behaviour of isoprene, total monoterpenoids, total sesquiterpenes, α-pinene, and limonene - separated into rainy and dry seasons - were added (Figs. A5-1 to A5-4).

*L350-351. Please discuss further on the claim of limonene's seasonality.*

Thank you for your clarifying question. We do not claim that the emission factor of limonene is seasonally dependent, but we write that there might be a possibility that it is (L348-351 in the original submission). We do not have more data, than presented in the original submission of the manuscript, to further explore a possible seasonality of the emission factor of limonene. The most likely reason for a possible seasonality is - and this is again speculation - changes in phenological condition and status of emitting species in the footprint area (see e.g. Fig. 2d-e). But as stated on L363-365 in the original submission of the manuscript - to our knowledge - emission rates have not been reported from the plant species in the footprint area, and thus we cannot prove this hypothesis. We will reformulate L350-351 in the original submission to "Thus, our results suggest that the EF for limonene might be seasonally dependent, which could be caused by changes in the phenology of emitting species in the footprint area, though we do not have any emission rate observations of the species to backup this hypothesis."

*Figure 7 contains very interesting information that needs to be better investigated and discussed (especially for the species that their emissions do not match with the results produced by the MEGAN equations).*

Thanks for your interest in these results. Since the lifetime of monoterpenes is a few hours (see Sec. 3.2), it is likely that part of the detected monoterpenes have been transported to the site from areas covered by other plant functional types than warm C4 grass and Crop1, such as broadleaved trees and shrubs, which are thought to have a significantly higher potential to emit monoterpenes (Guenther et al., 2012). It is, however, noteworthy that our estimated EF for β-pinene is in line with the listed value by Guenther et al. (2012) for warm C4 grass and Crop1, but not for broadleaved trees and shrubs, though the lifetime of β-pinene is within the same range as that of the other monoterpenes. MBO has a lifetime of about half a day, and thus a great part of the detected MBO does not originate from the near vicinity of the site, but can have been transported far. However, the EF listed in Guenther et al. (2012) for MBO for all plant functional types present in the relevant parts of Africa (https://doi.org/10.5194/gmd-5-1341-2012) is still about 2-3 orders of magnitude lower than estimated here. This might call for a revision of EFs for MBO, considering that also Jaars et al. (2016) found even higher concentrations of MBO than we did in this study, in an area of Africa which also should not contain MBO emitting species. We will add this extended discussion to the existing discussion in Sec. 3.3.

---

## Author Comment (AC2)

We appreciate your comments and helpful suggestions. We have added more information related to the sample collection, chemical analysis and the inverse model in our manuscript, the tables and figures were also revised in accordance with your and the Reviewer #2's suggestions. Our replies were addressed below.

**Anonymous Reviewer #1**

*This manuscript by Yang Liu and coauthors investigates the seasonal and diurnal variations of biogenic volatile organic compounds (BVOCs) ambient concentrations in southern Kenya. The authors focused on two contrasting ecosystems (highlands with agroforestry and lowlands with bushland and agriculture mosaic landscapes) during both the rainy and dry seasons. They report higher BVOC ambient concentrations in highlands vs. lowlands and during the rainy vs. dry season. The manuscript is very well written and will be suitable for publication after the authors address the following issues. I think that they do not go far enough in the discussion and do not really discuss nor explain the above-mentioned key results. In addition, key information is missing in the Methods section. Please find below detailed comments and suggestions that might help strengthen the manuscript.*

*General comments*

**1) Section 2.2:** *More detailed information is needed here regarding both the sampling and analytical methods.*
- *How were the samples stored and for how long (temperature and light conditions)?*

Thanks for pointing this out. The samples were stored in the freezer (at approximately-15°C) after collection (for 1 to 2 weeks) and before analysis (~ 2 months). Tubes were stored in a closed box with ambient temperature and dark inside, during the transportation to the Finnish Meteorology Institute (less than 1 week). This information has now been added to Section.2.2.

- *Did the authors check for potential losses of analytes during transport, storage, and chemical analysis? See for example Ortega and Helmig, 2008; Ortega et al., 2008; Angot et al., 2020.*

Thanks for this comment. We have found that the compounds are stable in our tubes for at least 2 months (Helin et al. 2020).

- *What was the temperature of the cartridges during sampling? A temperature > ambient temperature is generally used to prevent water accumulation on the adsorbent bed (Karbiwnyk et al., 2002).*

We used ambient temperature. However, sorbents used in the tubes were hydrophobic and therefore water was not accumulated. In addition, tubes were flushed with helium for 5 minutes with the flow of 50 ml/min before desorption and analysis to remove traces of humidity.

- *Please indicate the analytical uncertainty and detection limit for the compounds of interest?*

Thanks. The analytical uncertainties (U%) and limits of quantitation (LOQ) for studied compounds were shown in Appendix (Table A2). They were estimated as described in Helin et al. (2020).

**2) Section 2.5:** *More detailed information is also needed here to make this inverse modelling approach convincing.*
- *Please give the equations (rather than citing Guenther et al., 2012).*

Thank you very much for the suggestion. We followed it accordingly.

- *Describe how the activity factors were calculated. According to the Guenther et al. equations, one should use the average temperature and PFPD over the last 24 and 240 hours. Is that what you did? How about the use of leaf temperature, did you use ambient temperature instead? How do these values compare in the studied environment?*

True, we used temperature and PPFD averaged over the last 24 and 240 hours. We followed the reviewer's suggestion from above and added the specific equations and then it should become even clearer that the 24 and 240 hours averaged parameters were used. On Line 239 it is stated that we used leaf temperature, and that the leaf temperatures were calculated from observed air temperatures using Eqs 14.2 to 14.6 in Campbell et al. (1998).

- *I would like to see a discussion on model performances with figures/tables (this can be in the supplement). How do predicted and observed mixing ratios compare? This is absolutely essential; I won't trust the inverse modelling results without this additional section.*

We have attached a relevant table and two figures in Appendix (Table A5 and Fig A1-1 and A1-2).

**3) Lines 231-246 and Figure 4**: *in the text, the key message is the difference between the dry and rainy seasons. However, the way results are presented makes it easier to compare the two ecosystems rather than the two seasons. I would suggest reorganizing the structure of several sentences to show the difference between dry and rainy instead of highlands and lowlands (if that's what's intended). Example below:*

*"(…) the daily mean mixing ratio was higher during the rainy season than during the dry season. The daily mean isoprene mixing ratios ranged from 134 to 442 pptv in the highlands and from 22 to 69 pptv in the lowlands in the rainy season. (…) During the dry season, the isoprene mixing ratio ranged from 36 to 150 pptv in the highlands and from 6 to 15 pptv in the lowlands".*

*Could be transformed into:*

*"(…) the daily mean mixing ratio was higher during the rainy season than during the dry season. In the highlands, the daily mean isoprene mixing ratios ranged from 134 to 442 pptv in the rainy season vs. 36 to 150 pptv in the dry season. In the lowlands, the daily mean isoprene mixing ratios ranged from 22 to 69 pptv in the rainy season vs. 6 to 15 pptv in the lowlands".*

Thanks for improving the structure. The above text reorganized as you suggested (Line 267 to 273).

**4)** Same comment for **lines 239-246.**

Sentence modified (Line 275 to 282)

**5)** Same comment for **Figure 4**: *this figure shows the difference between highland and lowland while the discussion focuses on the difference between rainy and dry seasons. The scale is different on the two panels making it really difficult to compare the results. I do not really like this figure (see same comment below on Fig. 3), really hard to read. How about a boxplot instead if you do not really care about showing the temporal variability (that is not discussed anyway)? Something like that:*

*Panel a) isoprene, b) MTs, c) SQTs.*

*For each panel: y-axis is mixing ratio, x-axis shows boxplot for (1) rainy highland, (2) dry highland, (3) rainy lowland, (4) dry lowland.*

Thanks for improving this figure. Figure 4 replaced by a boxplots accordance with your suggestion, and the original figure moved to Fig. A2.

**6) Lines 250-253:** *I do not think that **Fig. 5** clearly describes what is discussed here. I would like to see an additional figure describing the diurnal cycle (something like Fig. 8 in Angot et al., 2020). In addition, is this diurnal cycle in line with that of environmental conditions (e.g., ambient temperature, light)?*

Thanks for the suggestion and for sharing an example to us. The Figure 5 modified follows your another comment about this Fig. 5. We added additional figures in Appendix A (Fig A5-1 to A5-4), which show the diurnal variation of compounds / compound groups, temperature and light. We also added the Fig. A3 to show the correlation coefficients between BVOCs and environmental factors.

As mentioned on L250-252 in the original submission, the mixing ratio of isoprene showed a distinct diurnal cycle in the highlands during both the rainy and dry seasons, but in the lowlands only during the dry season. The mixing ratio of isoprene increased in the morning, coinciding with sunrise, and stayed high during the rest of the day. As mentioned on L254-259 in the original submission, the mixing ratio of monoterpenes generally showed an opposite daily pattern with a bit higher concentrations during dark hours than light hours. These patterns are now also clearly illustrated in Figs. A5-1 to A5-4.

**7) Lines 263-266:** *Hard to see this on Fig. 5. I would convert panels c to f from mixing ratios to % contribution (similar to panels a and b). I would add the diurnal cycle of the various species on the additional figure discussed above. For this additional "diurnal cycle" figure, I suggest the following:*
*One panel per species: a) isoprene, b) MTs, c) SQTs, d) limonene, e) alpha pinene. Then 4 different colors per panel showing (1) rainy highland, (2) dry highland, (3) rainy lowland, (4) dry lowland.*

The subfigures from (c) to (f) have been converted from mixing ratios to percentage contribution, and those additional figures have been added to the Appendix A (Fig A5-1 to A5-4).

**8)** *In addition, I think the manuscript does not really discuss/explain why:*
- *Ambient concentrations are higher during the rainy season. Is it due to different humidity or ambient temperature?*

Thanks for this comment. We added the relevant discussion on lines 286 to 302. To summarize here: The higher ambient temperature, higher LAI, and lower mixing layer heights were likely the dominant factors which promoted the higher mixing ratios during the rainy season. Relative humidity and estimated atmospheric oxidant concentrations stayed largely the same during the both seasons in both ecosystems.

- *Ambient concentrations are higher in the highlands. Is it due to different tree species or different humidity/temperature?*
*These hypotheses can (and should) be tested using the model*

Thanks for your comment. A higher LAI and different plant species are very likely the dominant factors. This has now been pointed out in Sec. 3.1.1 (lines 286 to 302). The difference is not due to temperature, since the temperature is normally higher in the lowlands than in the highlands. Unfortunately, we are not able to test such hypotheses with the model, since we do not have sufficient input data to run the model for the highland site. This is also the reason why EF estimations were only conducted for Maktau in the original submission (Sec. 3.3). The highland site did not have the needed sensors to observe photosynthetic photon flux density nor soil moisture. Additionally, there was no flux tower at the highland site from which we could calculate mixing layer heights.

**9) Emission factors:**
- *According to Fig. 7, isoprene emissions are not seasonally dependent. In that case, what's driving higher concentrations during the rainy vs. dry season?*

Thank you for your clarifying question. Fig. 7a depicts the estimated emission factor (EF) for isoprene and not the predicted emission rate of isoprene nor the predicted canopy emission of isoprene, and thus we do not in

the figure, nor in the text, make claims about the seasonal dependency of the emission rate of isoprene nor the total canopy emission of isoprene – only about the emission factor for isoprene. The EF for isoprene is estimated to be less during the rainy season than during the dry season (though the difference might not be statistically different as mentioned in the manuscript), but the canopy emission of isoprene is higher during the rainy season than during the dry season, because the temperature (Fig. A4a, Table 1) and LAI (Rainy season: 2.08 in the highlands and 1.9 in the lowlands, Dry season: 1.53 in the highlands and 0.3 in the lowlands, Table 1) are significantly higher during the rainy season. Both promote higher canopy emissions. PPFD is higher during the dry season than during the rainy season (Fig. A4e), but the light conditions during the rainy season are still higher than the saturation point for the production and emission of isoprene (Fig. A4), and thus the slightly different light conditions between the two seasons do not impact the emission of isoprene. Additionally, the soil moisture is so low during the dry season that it reduces the emission rate of isoprene (Fig. A4c). Since the concentration of atmospheric oxidants stays largely the same during the two seasons, and since the boundary layer height is additionally a bit higher during the dry season (Fig. A4d), the concentration of isoprene is highest during the rainy season. So, to summarise: the higher ambient concentrations of isoprene during the rainy season are caused by higher canopy emissions of isoprene, which are mainly driven by higher temperatures, higher LAI, and an absence of dry soil moisture conditions. An elaboration was added to Sec. 3.1.1.

- *Since limonene is the only species showing a seasonal behavior, why don't you explore dependency on parameters such as light, temperature, soil moisture? (This is only done for isoprene and soil moisture).*

The emission factors for limonene, isoprene, as well as other compounds we presented have been modelled considering their dependency on light and temperature according to Guenther et al. (2012). The EF for isoprene was also modelled considering its dependency on soil moisture, because it has been previously investigated and there is an established way to describe it (Guenther et al., 2012). A dependency on soil moisture has not been considered in the calculation of the EF for limonene, because there is no established way to do that, since there do not exist many studies investigating the impact of soil moisture on the emission of monoterpenes. Moreover, we added an additional figure (Fig. A3) to show the correlation coefficients between BVOCs and environmental factors.

- *For isoprene, the authors investigated the influence of soil moisture. How about dependency on other factors such as light and temperature? This could help explain why concentrations are higher during the rainy vs. dry season.*

The emission factor for isoprene have been estimated considering the dependency of the emission of isoprene on light, temperature, and soil moisture as according to Guenther et al. (2012). About the higher mixing ratios during the rainy season, see our previous reply.

**Additional line-by-line comments**

**1) Line 84:** *Could you please add 1-2 sentence(s) in the introduction on why MBO is of particular interest?*

Thanks for this comment. The relevant sentence added on lines 87 to 90 and shows as follows:

'Although previous BVOC measurements detected small quantities of MBO from African ecosystems (Jaars et al., 2016; Liu et al., 2021), MBO oxidation is an important source of ozone and hydrogen radicals (Steiner et al., 2007), which are both important oxidants for new particular formation in the local atmosphere (Jaoui et al., 2012; Zhang et al., 2014).'

**2) Line 91:** *replace "considerable" by "significant".*

Thanks. The word was changed according to your suggestion.

**3) Line 92:** *Typo "We interested in the diurnal…".*

Thanks. We corrected this sentence.

**4) Section 2.1:** *One field campaign was performed in April, i.e., during the long rains, and another one in September, i.e., during the cool and dry season. The authors should perhaps mention the fact that no measurements were performed during the hot and dry season (Jan-Feb). There might be significant differences in BVOC ambient concentrations during the two dry seasons because of the difference in ambient temperature.*

Thanks for your comment. We added 'hot and long rainy season' and 'cool and long dry season' on lines 142 to 144. And we also added the sentence you mentioned about the 'hot and dry season' in Line: 375-377, which shows as follows:

'Be aware that no measurements were conducted during the short hot (January to February) and short cool (October to December) season, and it is likely that the mixing ratios of BVOCs are different during those seasons than what is presented here, due to differences in e.g. environmental conditions and phenology status.'

**5) Figure 3:** *This figure is really hard to read and thus not really useful as is. At minimum, please use the same y-axis during the rainy and dry seasons for easier comparison or provide a Table with mean values over the 4 campaigns. Another approach could be the use of boxplots (see comment on Fig. 4).*

Thanks for this suggestion. We replaced it with a boxplot and moved the original figure to Appendix (Fig. A4).

**6) Line 180:** *"O3 column densities to estimate surface (?) O3 concentrations".*

You're right, it is surface height. This information has now been added to the sentence.

**7) Lines 261-262:** *"MTs thus dominated the total BVOC mixing ratio during nighttime". Is this due to higher MTs at night or to a sharp isoprene diurnal cycle with low concentrations at night?*

Thank you for your question. According to our measurements, the mixing ratio of isoprene sharply decreases and stays low during night (Fig A5-1 to A5-4), and this behaviour is the main reason for MTs to dominate the total BVOC mixing ratio during dark hours. The mixing ratio of MTs is additionally slightly higher during dark hours than light hours.

**8) Lines 281-291:** *okay but why?*

Thanks for pointing this out. There are several factors affecting the concentration levels, e.g. dominant plant species and their distribution, temperature and light, wind speed/direction, mixing layer height, etc. But based on the limited details from the other sites, we were not able to find the key reasons for the differences in the concentration levels (lines: 345 to 348). However, we mentioned some possible reasons in the original submission, e.g. vegetation types, measured seasons.

**9) Table 1:** *For each species and study, please add mean or median value in parenthesis.*

Thanks for the suggestion. We added the mean and/or median values if the authors mentioned the values in their text (Table 2).

**10) Line 393:** *why not in the highlands as well?*

Thanks for your clarifying question. As also mentioned in our reply above, we did not have sufficient input data to run the model for the highland site (lacking PPFD, soil moisture, and mixing layer height). In the very end of Sec.2.5, we have now added the reason for not doing model simulations for the highland site.

References

Campbell, G. S., and Norman, J. M.: An Introduction to Environmental Biophysics, Springer, New York, 1998.

Guenther, A. B., Jiang, X., Heald, C. L., Sakulyanontvittaya, T., Duhl, T., Emmons, L. K., and Wang, X.: The Model of Emissions of Gases and Aerosols from Nature version 2.1 (MEGAN2.1): an extended and updated framework for modeling biogenic emissions, Geosci. Model Dev., 5, 1471–1492, https://doi.org/10.5194/gmd-5-1471-2012, 2012.

Jaars, K., Van Zyl, P. G., Beukes, J. P., Hellén, H., Vakkari, V., Josipovic, M., Venter, A. D., Räsänen, M., Knoetze, L., Cilliers D. P., Siebert, S. J., Kulmala, M., Rinne, J., Guenther, A., Laakso, L., and Hakola, H.: Measurements of biogenic volatile organic compounds at a grazed savannah grassland agricultural landscape in South Africa, Atmos. Chem. Phys., 16(24), 15665–15688, https://doi.org/10.5194/acp–16–15665–2016, 2016.

Jaoui, M., Kleindienst, T. E., Offenberg, J. H., Lewandowski, M., and Lonneman, W. A.: SOA formation from the atmospheric oxidation of 2-methyl-3-buten-2-ol and its implications for PM2.5, Atmos. Chem. Phys., 12, 2173–2188, https://doi.org/10.5194/acp-12-2173-2012, 2012.

Liu, Y., Schallhart, S., Tykkä, T., Räsänen, M., Merbold, L., Hellén, H., and Pellikka, P.: Biogenic volatile organic compounds in different ecosystems in southern Kenya. Atmos. Environ., 246, 118064, https://doi.org/10.1016/j.atmosenv.2020.118064, 2021.

Steiner, A. L., Tonse, S., Cohen, R. C., Goldstein, A. H., and Harley, R. A. Biogenic 2-methyl-3-buten-2-ol increases regional ozone and HOx sources, Geophys. Res. Lett., 34, L15806, doi:10.1029/2007GL030802, 2007.

Zhang, H. F., Zhang, Z. F., Cui, T. Q., Lin, Y. H., Bhathela, N. A., Ortega, J., Worton, D. R., Goldstein, A. H., Guenther, A., Jimenez, J. L., Gold, A., and Surratt, J. D.: Secondary Organic Aerosol Formation via 2-Methyl-3-buten-2-ol Photooxidation: Evidence of Acid-Catalyzed Reactive Uptake of Epoxides. Environmental Science & Technology Letters, 1 (4), 242-247, DOI: 10.1021/ez500055f, 2014.